# A Real-Time Incremental Video Mosaic Framework for UAV Remote Sensing

Ronghao Li [1], Pengqi Gao [2], Xiangyuan Cai [1], Xiaotong Chen [1], Jiangnan Wei [1], Yinqian Cheng [3] and Hongying Zhao [1,*]

1   School of Earth and Space Sciences, Peking University, Beijing 100871, China
2   National Astronomical Observatories, Chinese Academy of Sciences, Beijing 100864, China
3   Information Network Center, China University of Geosciences, Beijing 100083, China
*   Correspondence: zhaohy@pku.edu.cn

**Abstract:** Unmanned aerial vehicles (UAVs) are becoming increasingly popular in various fields such as agriculture, forest protection, resource exploration, and so on, due to their ability to capture high-resolution images quickly and efficiently at low altitudes. However, real-time image mosaicking of UAV image sequences, especially during long multi-strip flights, remains challenging. In this paper, a real-time incremental UAV image mosaicking framework is proposed, which only uses the UAV image sequence, and does not rely on global positioning system (GPS), ground control points (CGPs), or other auxiliary information. Our framework aims to reduce spatial distortion, increase the speed of the operation in the mosaicking process, and output high-quality panorama. To achieve this goal, we employ several strategies. First, the framework estimates the approximate position of each newly added frame and selects keyframes to improve efficiency. Then, the matching relationship between keyframes and other frames is obtained by using the estimated position. After that, a new optimization method based on minimizing weighted reprojection errors is adopted to carry out precise position calculation of the current frame, so as to reduce the deformation caused by cumulative errors. Finally, the weighted partition fusion method based on the Laplacian pyramid is used to fuse and update the local image in real time to achieve the best mosaic result. We have carried out a series of experiments which show that our system can output high-quality panorama in real time. The proposed keyframe selection strategy and local optimization strategy can minimize cumulative errors, the image fusion strategy is highly robust, and it can effectively improve the panorama quality.

**Keywords:** UAV remote sensing; image mosaicking; homography estimation; local optimization

## 1. Introduction

The increasing demand for high-resolution remote sensing images and basic geographic information across various sectors of society is driven by the development of the social economy and the need for national defense. This demand has become increasingly pressing and requires immediate attention. Moreover, the current demands for these products are progressively increasing regarding their level of quality. While satellite, airplane, and radar remote sensing data have their respective applications, in some cases they may not fully satisfy the needs of image data acquisition and processing. Compared to traditional aerospace remote sensing technology, UAV low-altitude remote sensing technology, as a new low-altitude remote sensing technology, has many advantages such as high flexibility, easy operation, high resolution, and low investment [1]. Most UAVs mainly work at a low altitude. As a result, the area that a single image can cover is small. Consequently, image mosaicking is an important technique for utilizing UAV multi-strip mage data. In general, image mosaicking involves various steps of processing: registration, reprojection, stitching, and blending [2]. According to registration algorithms, image mosaicking can be

divided into spatial domain-based algorithms and frequency domain-based algorithms. At the same time, some adopt the deep learning algorithms to do the image stitching end to end. In addition, many researchers have proposed new methods for UAV image mosaicking based on general image mosaicking. We will introduce each of these algorithms in the following sections.

### 1.1. Spatial Domain Image Mosaicking Algorithms

Image mosaicking algorithms based on spatial domains use pixel-to-pixel related information for registration. Most of the existing algorithms fall into this category. Spatial domain-based image mosaicking can be either area-based or feature-based. Area-based algorithms rely on computation between "windows" of pixel values in the two images, which need to be mosaicked [3]. For example, normalized cross correlation can be used as a metric to calculate the pixel distribution similarity of each window in the images [4]. Mutual Information can be used to calculate the shared information between two images to measure the similarity. However, due to the fact that area-based algorithms match images by comparing patches, these techniques have the disadvantage of being computationally slow, and require high areas of overlap between input images.

Different from area-based algorithms, feature-based algorithms do not require large areas of overlap between two images. Instead, they use pixel-level features to match images and calculate the geometric transformation between a pair of images. These algorithms rely on feature extraction algorithms, which can detect significant features in an image, including edges, corners, domain histograms, etc., and assign a descriptor to each feature for comparison with the features of another image. Depending on the types of features extracted, feature-based algorithms can be classified into low-level feature-based algorithms and contour-based algorithms. Low-level feature-based algorithms usually determine features and descriptors by calculating the distribution and gradients of the surrounding domain.

Scale invariant feature transform (SIFT) [5] is a feature detector and descriptor that is invariant to image scaling and rotation in most cases, and also has a tolerance to illumination and 3D camera viewpoint change, but this algorithm needs a large amount of computation. Based on SIFT, Speeded Up Robust Features (SURF) [6] is proposed. Similar to SIFT, this algorithm is also based on scale space. However, SURF uses the Hessian matrix of integrated images to estimate the local maximum values of different scale spaces. The oriented FAST and rotated BRIEF (ORB) [7] uses the feature from accelerated segment test (FAST) algorithm to detect feature points of the image, and then describes the detected feature points with the binary robust independent elementary feature (BRIEF) algorithm.

Since low-level features are not intuitive for human perception, contour features can be another choice, and they are high-level features because they are more natural to human perception. These high-level features focus mainly on extracting shape or texture in the image. Regions with different structures are described as different descriptors and matched in both images. This type of algorithm is suitable for complex parameters and motion models, as it looks for advanced features under extreme image changes [8–10].

### 1.2. Frequency Domain Image Mosaicking Algorithms

Unlike spatial domain-based image mosaicking algorithms, frequency domain image mosaicking algorithms require computation in the frequency domain to find the optimal transformation parameters between a pair of images. These algorithms adopt phase correlation properties to register images, usually using Fourier transform and inverse Fourier transform to transform between spatial and frequency domains and register images in the frequency domain. However, the frequency domain image mosaicking algorithms are too sensitive to noise, and can only get a rough result. To obtain accurate results, the overlap between the images needs to be high.

*1.3. Deep Learning-Based Image Mosaicking Algorithms*

With the rapid development of deep learning, many scholars have tried to use deep learning algorithms for image mosaicking tasks. There are two main development directions in this category. One is to learn features and descriptors of images to obtain features with stronger generalization capability than traditional handcrafted features, and the other attempts to use an end-to-end approach for image stitching.

SuperPoint [11] is a self-supervised framework for training interest point detectors and descriptors suitable for a large number of multiple-view geometry problems in computer vision, but it uses self-generated data to train for corner detection; its generalization for real scenarios needs to be verified. Learned invariant feature transform (LIFT) [12], a recently introduced convolutional replacement for SIFT, stays close to the traditional patch-based detect-then-describe recipe; the limited applicability of this algorithm is due to its dependence on image patches as input rather than entire images, which prevents the algorithm from processing the entirety of an image for mosaicking. L2-Net [13] proposes to learn high-performance descriptors in Euclidean space via the convolutional neural network (CNN); Unsuperpoint [14] utilizes a siamese network and a novel loss function that enables interest point scores and positions to be learned automatically using a self-supervised approach. Compared to handcrafted algorithms such as SIFT and SURF, the feature points and descriptors extracted by a neural network can express deeper features of images and have a stronger generalization ability. However, the correct key points that can be identified by these algorithms are limited. Under the condition that the number of key points is not limited, the handcrafted algorithms can obtain more accurate matching through quantitative advantage.

Some scholars consider using deep neural networks (DNN) to learn the transformation relationship between image pairs. Geometric Matching Networks (GMN) [15] and Deep Image Homography Estimation (DIHE) [16] use a similar self-supervision strategy to create training data for estimating global transformations. Some algorithms use an individual homography estimation network for coarse alignment and optimize the pre-aligned images by reconstruction networks to achieve better stitching results in large-baseline scenes [17,18]. However, the current deep-learning-based algorithms can only input two images and output a panorama; the deep learning framework of multi-strip sequences remains to be studied.

*1.4. UAV Image Mosaicking Algorithms*

With the advancements in UAV and sensor technology, UAVs are being increasingly used for photogrammetric data acquisition. Their low flight altitude makes them ideal for capturing high-resolution images of small to medium-sized areas. However, when it comes to large areas, image mosaicking technology is needed to stitch together multiple images into a single panorama. Compared with other requirements, the UAV often obtains multi-strip and large-scale image sequences. In this case, it is difficult to obtain panoramas by directly stitching images. In general, image mosaicking for UAV missions focuses more on improving the performance and efficiency of multi-strip image sequences.

The high resolution and large number of images acquired by UAV leads to huge time consumption of image mosaicking. Yahyanejad [19] used multi-source data mixing to try to accelerate the stitching process. They determined the rough position of each image with the help of the GPS information or inertial measurement unit data carried by the UAV, and carried out feature matching in the rough position to reduce the retrieval space of image features. This algorithm can obtain panoramic images without obvious distortion of the view angle, and retain certain geo-reference information from the GPS. However, this algorithm did not solve the cumulative error in a large sequence of image mosaicking. Danilo Avola [20] proposed an algorithm to obtain panoramic images increasingly at low altitudes. In order to accelerate matching, they adopted A-KAZE features instead of SIFT and ORB features for feature matching. They also used ROI to reduce the amount of calculation for each new frame and adopted rigid transformation to replace homography.

However, the rigid transformation did not perform well with complex terrain. Liu [21] proposed an integrated GPS/INS/Vision system. They assumed a negligible change in ground height between two adjacent frames during the UAV aerial mission. After obtaining the image, GPS/IMU was used for geometric correction, and then the transformation of image pixels was interpreted as a linear function transformation, and a parameter was added for places with large terrain fluctuations. Compared with other algorithms, their operation speed was faster, but as their algorithm focuses on raising the accuracy of the corresponding points, the image stitching errors were obvious even though the accuracy of corresponding points was high.

The other group focuses on how to reduce the accumulated errors in UAV image mosaicking. Zhang [1] proposed an optimization algorithm. They introduced the Levenberg–Marquardt (LM) algorithm to globally optimize the position of an image in the panorama and generate a panorama with high precision, but their global optimization does not allow incremental input. Zhao [22] presented an online sequential orthophoto mosaicking solution for large baseline high-resolution aerial images with high efficiency and novel precision. An appearance- and spatial-correlation-constrained fast low-overlap neighbor candidate query and matching strategy were used for efficient and robust global matching, but this algorithm requires a very high altitude of UAV. Ren [23] proposed a simplified algorithm for UAV stitching. Based on image contrast, they determined the optimal band for extracting SIFT, and then extracted SIFT on a single band image to improve the speed. A simplified projection model was proposed to avoid huge computation caused by 3D reconstruction and irregular resampling. Jyun-Gu [24] proposed a novel speed estimation algorithm capable of measuring the distance of pixel movement between consecutive frames. However, this algorithm was limited in that the flight path must be a straight line, and that moving objects in the scene would affect the algorithm's estimation of the pixel motion speed, resulting in distortion of the panorama. Chen [25] proposed a nonrigid matching algorithm based on motion field interpolation; the homography transformation relationship was improved by vector field consensus (VFC), which had better robustness to image mosaicking, but required a lot of computation. Map2DFusion [26] proposed a real-time approach to stitch large-scale aerial images incrementally. It used a monocular SLAM system to estimate camera position and attitude, but this algorithm relies on ORBSLAM or other SLAM systems. Zhang [27] proposed a novel image-only real-time UAV image mosaic framework for long-distance multistrip flights, and it does not require any auxiliary information such as GPS or GCPs; their optimize algorithm is worth learning, and this algorithm mitigates the cumulative error by using a least-squares-based pose optimization, but it is too sensitive to noise.

Although considerable progress has been achieved in this area, a fast, robust, and efficient aerial image mosaic system in an unknown environment is still worthwhile to study, especially in long-distance multi-strip flights. There are two main problems in the current UAV image mosaic algorithm. First, most of the UAV image mosaicking algorithms are not real-time. They require all the image sequences before they can perform global optimization. The quality of the image obtained is relatively high. However, it cannot perform fast, real-time, and incremental mosaicking. It is difficult to achieve the expected effect in some tasks that need to quickly obtain the panorama of the region of interest, such as the task of simultaneous mapping in flight. In the algorithms that are in real time, most of them focus on fusing sensor data, reducing the matching space, and replacing the features to accelerate the algorithm. The advantage is that they can be stitched incrementally, but they cannot solve the error accumulation problem caused by the incremental stitching of long sequences of UAV images.

In this paper, a real-time UAV image mosaic framework is proposed, aimed at tasks such as rapid post-disaster reconstruction and search and rescue operations, which allows the UAV to synchronously return images and incrementally assemble them during the execution of aerial photography tasks. Core technologies used to construct our framework are:

1. A fuzzy positioning-based keyframe selection strategy, which greatly improves the efficiency of the algorithm without compromising the stitching effect.
2. A new local optimization strategy to minimize the weighted reprojection error, which minimizes the cumulative error when stitching sequences from multi-strip, which has a large number of images.
3. A partition-weighted pyramid fusion algorithm, which is used to give the best visual effect to the generated panoramas.
4. Although our algorithm is image-only mosaicking if the GPS corresponding to the image can be obtained, our algorithm also supports generating a panorama with geographic coordinates.

## 2. Materials and Methods

Our framework consists of two simultaneous branches: the data synchronization branch and the image mosaicking branch. The data synchronization branch is mainly responsible for real-time video stream acquisition and data synchronization with the image mosaicking branch during the flight of the UAV, while the image mosaicking branch splits the acquired video stream into frames, and stitches the acquired frames. Our main innovative work focuses on the calculation branch.

The flow of our algorithm mainly includes four parts:

1. Initialization of the mosaicking
2. Fuzzy location of new frame and keyframe selection;
3. Local pose optimization of keyframe
4. Expanding and generating panoramas.

The overall frame framework is shown in Figure 1.

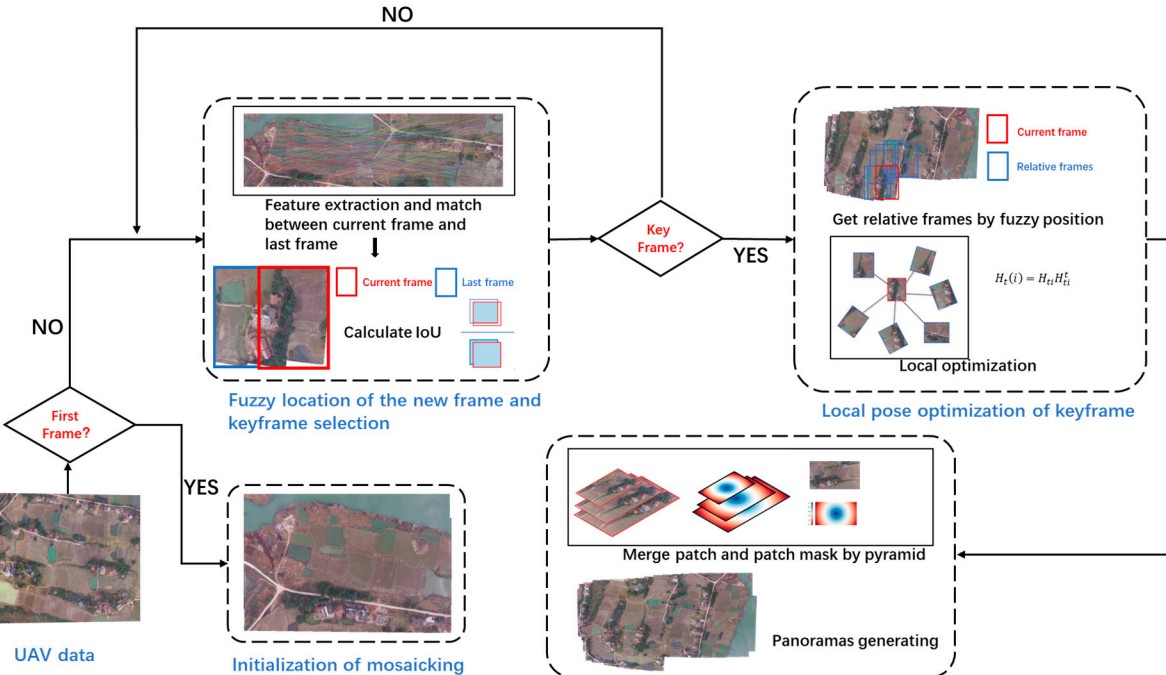

**Figure 1.** The overall framework. The framework mainly includes four tasks: (1) initialization of mosaicking, (2) fuzzy location of the new frame and keyframe selection, (3) local pose optimization of keyframe, (4) expanding and generating panoramas.

### 2.1. Initialization of Mosaicking

After the UAV starts flying, the data synchronization branch picks up video streaming data. Generally speaking, the camera's optical axis can be kept perpendicular to the ground with the help of the cradle head during flight. The aerial survey task of the UAV starts from

the flight to the specified height. When the first image is captured by the UAV, the system first carries out an automatic initialization process. We set the first frame as the base plane of other images, and subsequent images are mapped to this plane.

We calculate and record the flight trajectory and the fuzzy position of keyframes, which provide optimization information for the subsequent image mosaic. At the same time, we use SIFT to extract features in the frame. The experimental results show that SIFT extraction algorithms are the most balanced choice in UAV image mosaicking. In fact, our algorithm supports most handcrafted features. In order to ensure generalization, we can also replace SIFT with other deep-learning feature extraction algorithms.

### 2.2. Fuzzy Location of the New Frame and Keyframe Selection

In the stitching process, not all the frames are necessary. If the stitching is dense, it will lead to a rapid increase in information redundancy and computational burden, so it is important to select keyframes.

The newly added image is called the current frame $K_t$, and we will calculate its position relative to the previous keyframe $K_{t-1}$ by matching each SIFT feature descriptor $D_t(x_i, y_i)$ of $K_t$ with the descriptor $D_{t-1}(x, y)$ of $K_{t-1}$. $D_j(x_i, y_i)$ represents the descriptor of SIFT keypoints at coordinate $(x_i, y_i)$ in frame $K_j$, which is described as a 128-dimensional vector [5]:

$$D_j(x_i, y_i) = [\theta_1, \theta_2, \dots, \theta_{128}] \tag{1}$$

The feature descriptor of SIFT extraction algorithms is composed of the local gradient information calculated in the region around the keypoint. This gradient information is obtained by statistics of gradient direction histograms in a specific way, then the histograms are concatenated into a 128-dimensional vector as the descriptor of the keypoint. The SIFT feature descriptor is characterized by good invariance to image rotation and scaling. We find two neighboring matches, the nearest neighbor $(D_t(x_i, y_i), D_{t-1}(x_k, y_k))$ and the next nearest neighbor $(D_t(x_i, y_i), D_{t-1}(x_j, y_j))$. In order to maintain robustness, false matches should be eliminated. It is generally assumed that if two pairs of neighboring matches have similar internal distances, they have a high probability of being false matches, and a threshold $P$ is used to filter which matches are correct.

$$\|D_t(x_i, y_i) - D_{t-1}(x_k, y_k)\|_2 < P\|D_t(x_i, y_i) - D_{t-1}(x_j, y_j)\|_2 \tag{2}$$

After matching the feature points of two images, we need to calculate the relative position between the two images with the matched feature points. While the scenes are almost planar, the homography can precisely describe the projective transformation that relates to two images of the same scene [28]. In this paper, the homography model is used to describe the inter-image projection. A homography that relates two planes is usually represented as an invertible $3 \times 3$ matrix [29].

$$H = \begin{bmatrix} h_1 & h_2 & h_3 \\ h_4 & h_5 & h_6 \\ h_7 & h_8 & 1 \end{bmatrix} \tag{3}$$

It is constrained by 8 independent parameters, requiring at least 4 corresponding points for determination. We use random sample consensus (RANSAC) to estimate the homography $H_t^{t-1}$ from the current frame $K_t$ to the previous frame $K_{t-1}$. The homography transforms the pixel coordinates $[x, y, 1]$ in the current frame to that in the previous one.

$$\begin{bmatrix} x_{t-1} \\ y_{t-1} \\ 1 \end{bmatrix} = H_t^{t-1} \begin{bmatrix} x_t \\ y_t \\ 1 \end{bmatrix} \tag{4}$$

The homography transforms the pixel coordinates in the previous frame $K_{t-1}$ to that in the base plane $K_{base}$, which can be described as the following formula:

$$\begin{bmatrix} x_{base} \\ y_{base} \\ 1 \end{bmatrix} = H_{t-1} \begin{bmatrix} x_{t-1} \\ y_{t-1} \\ 1 \end{bmatrix} \tag{5}$$

where $(x_t, y_t)$ denotes the coordinates of the image point in the current frame, $(x_{t-1}, y_{t-1})$ denotes the coordinates of the corresponding point in the previous frame, and e$H_{t-1}$ denotes the homography from the previous frame $K_{t-1}$ to the base plane. The projection from the current frame to the base plane can be calculated using the homography $H'_t = H_{t-1} H_t^{t-1}$. Due to accumulated error, the position calculated by this homography matrix is not very accurate. We call it a fuzzy position. The real position of the current frame should be close to this fuzzy position. As shown in Figure 2, the minimum bounding rectangle of the position contour is used to describe the fuzzy position, and it is assumed that the real position of the frame should be located in this rectangle.

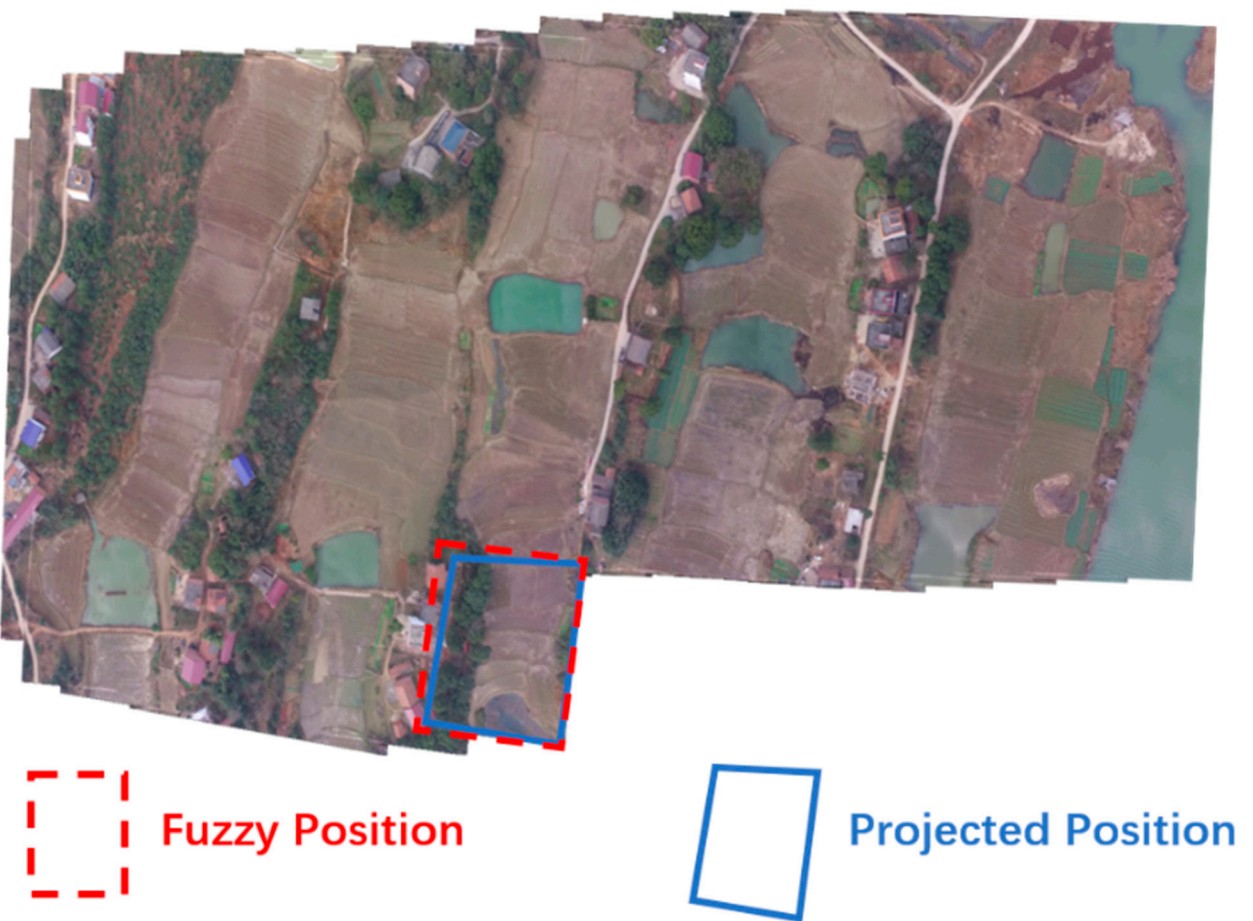

**Figure 2.** Fuzzy location. Red dashed box represents fuzzy position and blue box represents projected position. Fuzzy position is the minimum bounding rectangle of projected position.

To compromise between speed and mosaic quality, a specified keyframe selection strategy is proposed in the framework. If the keyframes are sparse, the overlap between frames is too small, resulting in stitching failure. If the keyframes are dense, there will be a lot of redundant information. Such information cannot improve the results much, but does cost a lot of computing resources. In this paper, the keyframe selection strategy is based on the intersection over union (*IoU*) between the fuzzy position of the current frame and the

position of the previous frame. We define the *IoU* between the current frame $K_t$ and the previous keyframe $K_{t-1}$ as shown in the following formula [30]:

$$IoU(t, t-1) = \frac{Area(K_t) \cap Area(K_{t-1})}{Area(K_t) \cup Area(K_{t-1})} \tag{6}$$

As shown in Figure 3, $Area(K)$ denotes the area of frame $K$ in the base plane. $\cap$ denotes the intersection between the two images, and $\cup$ denotes the union of the two images. The *IoU* value ranges from 0 to 1, with larger values indicating greater overlap between the two images and hence greater similarity, and smaller values indicating less overlap and hence less similarity.

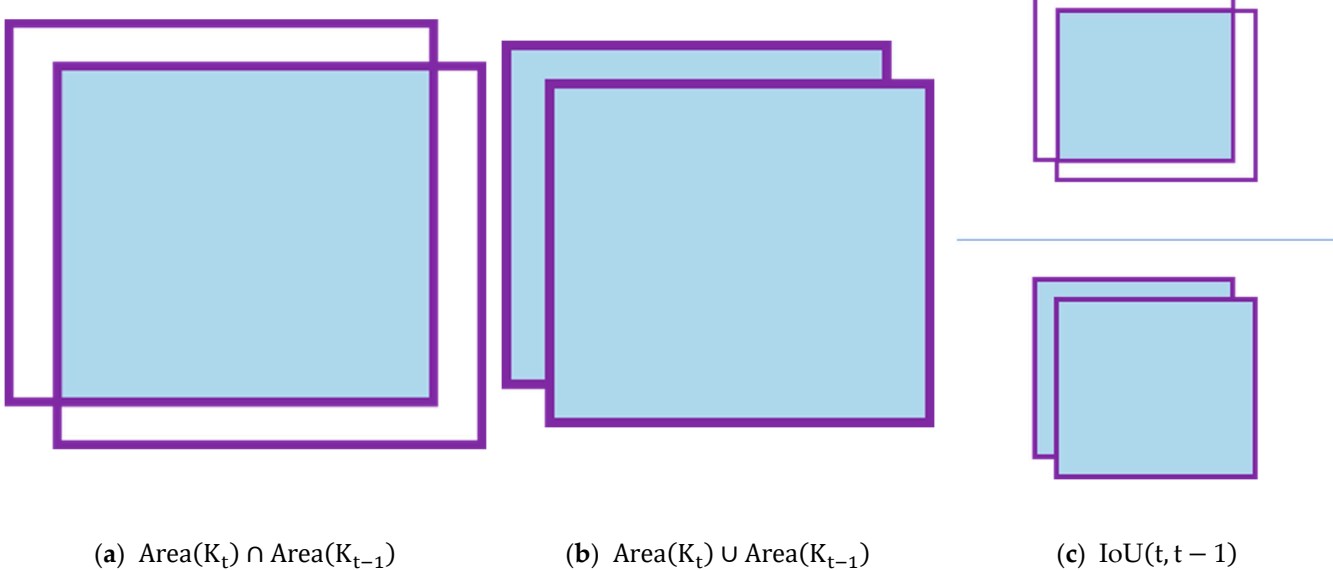

(**a**) Area($K_t$) $\cap$ Area($K_{t-1}$)  (**b**) Area($K_t$) $\cup$ Area($K_{t-1}$)  (**c**) IoU(t, t − 1)

**Figure 3.** *IoU* of two areas. *IoU* is calculated by dividing the overlap between two areas by the union of these.

Two thresholds $P_{\min}$ and $P_{max}$ are defined as low and high limits for keyframe determination. Compared with these thresholds, if $IoU(t, t-1) > P_{(max)}$, the overlapping area of two frames is too large, and we consider that the current frame does not introduce enough information; if $IoU(t, t-1) < P_{min}$, we consider that the overlapping area of two frames is too small, meaning it is easy to introduce noise and get the wrong match and transformation; if $P_{min} \leq IoU \leq P_{max}$, the current frame is considered a keyframe.

## 2.3. Local Pose Optimization of Keyframe

For incremental image mosaicking, the naive idea of calculating the position of the current frame $K_t$ is to calculate the homography matrix $H_t^{t-1}$, the homography $H_{t-1}$, and then project $K_t$ to the base plane. We can derive this relationship from Formulas (4) and (5)

$$H_t = H_{t-1} H_t^{t-1} \tag{7}$$

This formula represents the candidate homography when the current frame matches $K_{ti}$. We need to choose the best of the $t - 1$ candidate homographies that we calculated. Although this approach is intuitive and simple, it can cause very large cumulative errors. Li [31] analyzed the cumulative error of the sequence images. They subdivided the cumulative error of matrix calculation into addition errors and multiplication errors. It was found that the error propagation in matrix operation is very fast. This error leads to worse mosaicking results in later stages.

We propose a new local optimization strategy to mitigate cumulative errors. When the current frame is determined as keyframe $K_t$, we not only calculate the position from the previous frame $K_{t-1}$, but also try to obtain more information to optimize the position of $K_t$.

As mentioned above, we store the position of every keyframe $K_i(i < t)$. We need to quickly determine the relative keyframes that match the current frame $K_t$. The *IoU* between the current frame $K_t$ and every previous keyframe is calculated. If the *IoU* between $K_t$ and a frame $K_i$ is greater than a predefined threshold $P$, $K_i$ is considered the relative frame of the current frame. All such frames consist of a candidate keyframe set. The frames in this set will provide important information to optimize the position of the current frame.

$$S = \{K_m | IoU(K_m, K_t) < P, 0 \le m < t\} \tag{8}$$

In this formula, we create a set that contains all frames for which the *IoU* satisfies the condition. After that, the transformations of the current frame from the frames in the set are calculated by matching the features of the pair. For each frame $K_{t1}$, $K_{t2}, \ldots, K_{tn}$ in the set, their homography matrices to the base plane are $H_{t1}, H_{t2}, \ldots, H_{tn}$. For any keyframe $K_{ti}$ in this set, we can get the homography $H_{ti}^t$ from $K_t$ to $K_{ti}$, then the candidate homography $H_t(i)$ from the current frame and the base plane can be calculated through the following formula, which has a form that is similar to Formula (7):

$$H_t(i) = H_{ti} H_{ti}^t \tag{9}$$

A good homography should be one with as little cumulative error as possible. We can calculate all the candidate homography and then query the optimal one. The weighted reprojection error is used as the criterion to judge whether the homography is optimal, which is defined as follows:

$$E(i) = \sum_{j \neq i}^{n} \frac{1}{IoU(t, tj)} \sum_{k \in C_{tj} \cap C_t} \sqrt{\left( Proj(x_{tj}^k, H_{tj}) - Proj(x_t^k, H_t(i)) \right)^2 + \left( Proj\left(y_{tj}^k, H_{tj}\right) - Proj(y_t^k, H_t(i)) \right)^2} \tag{10}$$

where $C_{tj} \cap C_t$ represents the matched feature points between $K_t$ and $K_{tj}$, and the function $Proj(x, H)$ represents the coordinate of the point after projecting it back to the reference plane through the homography $H$. We use *IoU* to balance out the difference in the number of feature points between different matched image pairs.

$$m = \underset{i}{\arg\min} E(i) \tag{11}$$

In Formula (11), m is the index of the best frame we find. Through Formula (10), we can determine the best frame $k_{tm}$ that matches the current frame, and the homography from $K_t$ to $K_{tm}$ is used to project the current frame to the base plane, as referred to in Formula (8). Then, the homography from $K_t$ to the base plane is determined.

$$H_t = H_t(m) \tag{12}$$

### 2.4. Expanding and Generating Panoramas

After local optimization, the keyframe can be projected to the base plane through the resulting homography. However, due to errors caused by image distortion and geometric offset, there will still be misalignment between two geometrically aligned images. We can use some fusion strategies to generate better panoramas. Other methods usually find the best stitching seam to obtain better results. However, the computational cost is very high. In our framework, we refer to the algorithm from map2dfusion [26]. We use a partition-weighted pyramid fusion algorithm to generate the stitching results naturally, and use a partition processing method to make the process faster.

In order to reduce the exposure differences and misalignments between the images, a Gaussian pyramid is constructed to fuse multiple scale spaces, and a Laplacian pyramid

is computed in the construction process for the restoration of the original image with an expanded operation [32].

To quicken the operation, a k-level Gaussian pyramid is computed first, and each level is subtracted from the lower level of the pyramid [33]:

$$G_l(x,y) = \sum_{d_x=-n}^{n} \sum_{d_y=-n}^{n} w_{d_x,d_y} G_{l-1}(x + d_x, y + d_y) \tag{13}$$

$$L_l = G_l - G_{(l+1)} \, l < k \tag{14}$$

The highest level equals since there is no higher level computed.

The Gaussian pyramids of the two images have the overlapping area. In traditional methods, the pixel of the overlapping area is usually obtained by the weighted sum or average of the pixel of the two pyramid layers. However, this approach will inevitably produce blur and ghosting.

For an aerial image, the center part of the image is more ortho and has both less distortion and a smaller scale change than the edge. For each image, we construct a mask representing how close each pixel is to the center of the image. As shown in Figure 4, for an image of size $w \times h$, we can build a mask $M$ of the same size. The pixel value in the mask is as follows:

$$M(x,y) = 1 - \frac{\sqrt{\left(x - \frac{w}{2}\right)^2 + \left(y - \frac{h}{2}\right)}}{\sqrt{\left(\frac{w}{2}\right)^2 + \left(\frac{h}{2}\right)^2}} \tag{15}$$

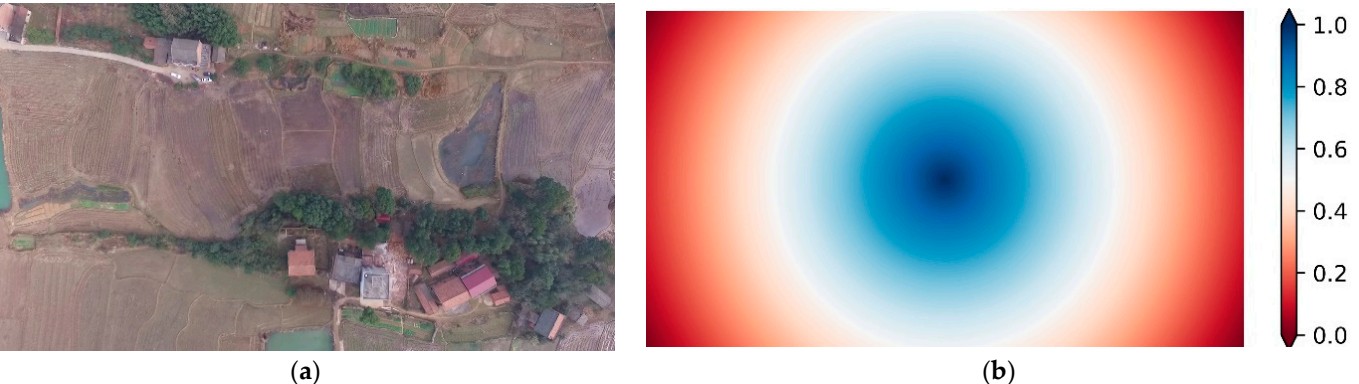

**(a)**       **(b)**

**Figure 4.** An image and its corresponding mask. (**a**) Image. (**b**) The mask of this image. The value in the mask represents how close each pixel is to the center of the image.

As you can see from this formula, we construct a mask with decreasing values from the center of the image to the edge. This formula is used to generate a mask for an image, where the value of the mask falls within the range of 0 to 1. The value of each pixel in the formula depends on its distance to the center of the image and the diagonal length of the image. The farther the distance, the smaller the corresponding mask value, and the closer the distance, the larger the corresponding mask value. The mask will do the same homography transformation as the corresponding image. The pixels in the overlapping part of the pyramid are determined by comparing the mask pixel at the same position.

$$K_{map}(x,y) = I(x,y)K_{map}(x,y) + (1 - I(x,y))K_t(x,y) \tag{16}$$

$$I(x,y) = \begin{cases} 1 & M_{map}(x,y) > M_t(x,y) \\ 0 & M_{map}(x,y) \leq M_t(x,y) \end{cases} \tag{17}$$

At the beginning of image fusion, we initialize a canvas of fixed size. When the boundary of the new frame exceeds the canvas boundary, the boundary will be dynamically expanded to ensure the integrity of the canvas.

In the process of stitching, the panorama keeps growing, and in the fusion algorithm, it is very computationally expensive to build the pyramid and compare the mask size pixel by pixel. Therefore, we adopt the method of local fusion. We captured the pixel change area of the new frame and captured the corresponding area of the panorama and two masks. After fusing them with our algorithm, they are pasted back onto the original canvas. Thus, it greatly reduces the computational complexity and accelerates the operation speed.

*2.5. Generating Panorama Geographic Coordinates*

Our algorithm can incrementally output high-quality panoramas from image-only input. In some scenarios, we want to output the panoramic with geographic information. This scenario inevitably requires us to input some location information to assist in the stitching process. We can use a small amount of GPS at the beginning of the flight to obtain the geographic information of the panorama. For the first n frame keyframe $K_i$, we obtain the GPS coordinates $(lat_i, lng_i)$f the image center. We convert the GPS coordinates to the UTM coordinate system $(X_i, Y_i)$, and obtain the coordinates on the base plane $(x_i, y_i)$. We believe that in the case that the base plane is parallel to the ground, a simple model can be used to describe their transformation:

$$\begin{cases} X_i = \alpha\cos\theta x_i - \alpha\sin\theta y_i + C_1 \\ Y_i = \alpha\sin\theta x_i + \alpha\cos\theta y_i + C_2 \end{cases} \tag{18}$$

We use this formula to describe a simple rotation and translation model. Let $A = \cos\theta$, $B = \sin\theta$, and then it can be described as format of matrix operation:

$$\begin{bmatrix} x_i & -y_i & 1 & 0 \\ y_i & x_i & 0 & 1 \end{bmatrix} \begin{bmatrix} A \\ B \\ C_1 \\ C_2 \end{bmatrix} = \begin{bmatrix} X_i \\ Y_i \end{bmatrix} \tag{19}$$

When $n > 2$, we have some redundant observations, and we can construct the formula:

$$\begin{bmatrix} x_1 & -y_1 & 1 & 0 \\ y_1 & x_1 & 0 & 0 \\ x_2 & -y_2 & 1 & 0 \\ y_2 & x_2 & 0 & 1 \\ & & \vdots & \\ x_n & -y_n & 1 & 0 \\ y_n & x_n & 0 & 1 \end{bmatrix} \begin{bmatrix} A \\ B \\ C_1 \\ C_2 \end{bmatrix} = \begin{bmatrix} X_1 \\ Y_1 \\ X_2 \\ X_2 \\ \vdots \\ X_n \\ X_n \end{bmatrix} \tag{20}$$

We transform the problem into an overdetermined equation to solve the problem. We can use QR decomposition to quickly solve the overdetermined equation and obtain four parameters.

For the following keyframes, we can calculate the coordinates of the corner points of the image in the base plane so that the panoramic image with geographic information can be generated.

## 3. Experiment and Result

In this section, we evaluate our algorithm through a series of experiments. First, we test the overall feasibility and performance of our algorithm with the most popular datasets and our data. Then, we verify the environmental adaptability and robustness of the algorithm by simulating changes in real-world scenarios and sensor variations, such as changes in brightness and random noise.

We also verify the algorithm selection of each strategy in our framework. First, we prove the excellent performance of SIFT by testing the performance and robustness of various feature extraction algorithms under the influence of geometric transformations, brightness changes, and random noise. Then, we verify that our frame extraction strategy greatly reduces the mosaic speed without affecting the mosaic results. Finally, we show that our local pose optimization algorithm and fusion algorithm have superior performance compared to the popular ones.

### 3.1. Dataset and Experimental Setup

To evaluate the effectiveness of the algorithm, we check it in real UAV aerial sequences. The NPU drone map dataset is adopted; this data set is available on the website [34]. It contains several aerial video sequences taken at different terrains and altitudes and is widely used to evaluate aerial image mosaics. This dataset was collected by the Phantom3 during flight. In addition to the publicly available dataset, we also collected some data using our UAV, the CW-15. Table 1 provides detailed description of the UAVs and sensor parameters used to acquire data.

**Table 1.** Detailed description of the UAVs and sensors parameters used to acquire data.

| UAV | Camera | CMOS Size (inch) | Focal Distance (mm) | FOV | Photo Size |
|---------|-----------------|------------------|---------------------|-------|--------------------|
| Phantom3 | Phantom3-Camera | 1/2.3 | 20 | 94° | 4000 × 3000 |
| CW-15 | CA-103 | 1/1.7 | 31.7 | 66.6° | 3840 × 2160 |

In order to measure the efficiency of our algorithm and reproduce our results more intuitively, we carefully documented our hardware configuration. Our experiments were conducted on a laptop running the Windows 11 operating system, equipped with a Ryzen 5600 U (CPU) and 16 GB RAM. The system configuration provided memory capacity for our experiments, allowing us to carry out large-scale image processing tasks efficiently. Our hardware configuration of the Ryzen 5600 U processor and 16 GB of RAM is a commonly found memory setup and is a low-voltage processor, which further emphasizes that our algorithm is not reliant on high-end hardware.

### 3.2. Real-Scene Experiment

This section describes a set of experiments designed to validate the performance of the proposed framework in real UAV aerial scenes. In the NPU drone map, we select three groups of photographic sequences, including three scenes: village, highway, and factory. We also took some photos with our own UAV and selected a complex industrial park for testing. See Table 2 for the detailed data of image sequences.

**Table 2.** Test Image Sequences Information. 'H-max' denotes the maximum flight height and "*Area*" denotes the area of ground.

| Sequence | Location | UAV | H-Max (m) | Area (km$^2$) | Frames |
|-------------------|-------------------------|----------|-----------|---------------|--------|
| Phantom3-village | Hengdong, Hunan | Phantom3 | 196.6 | 0.932 | 406 |
| Phantom3-huangqi | Hengdong, Hunan | Phantom3 | 222.3 | 1.313 | 393 |
| Phantom3-factory | Luoyang, Henan | Phantom3 | 181.8 | 0.782 | 406 |
| CW15-boyang | Gongqingcheng, Jiangxi | CW15 | 152.5 | 0.735 | 235 |

From Figure 5, we can see the experimental results of sequences in the Phantom3-factory, Phantom3-huangqi, and Phantom3-factory selected from the NPU drone map, and our own data CW15-boyang.

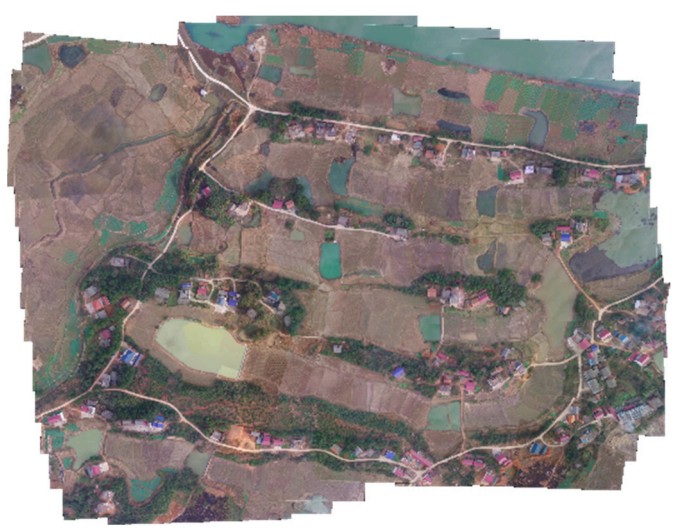

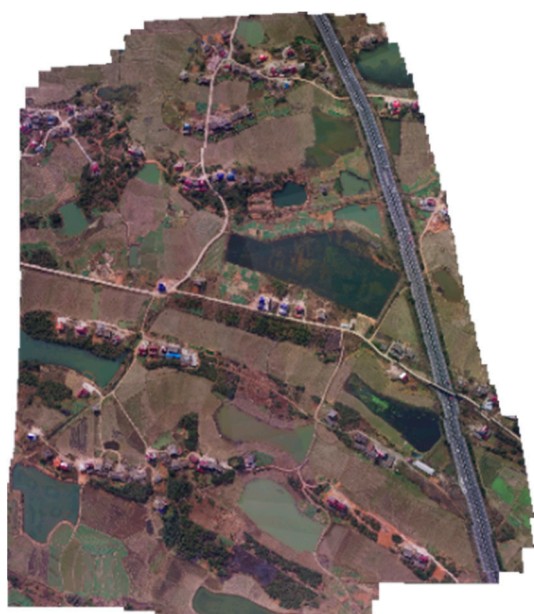

(**a**) Phantom3-village

(**b**) Phantom3-huangqi

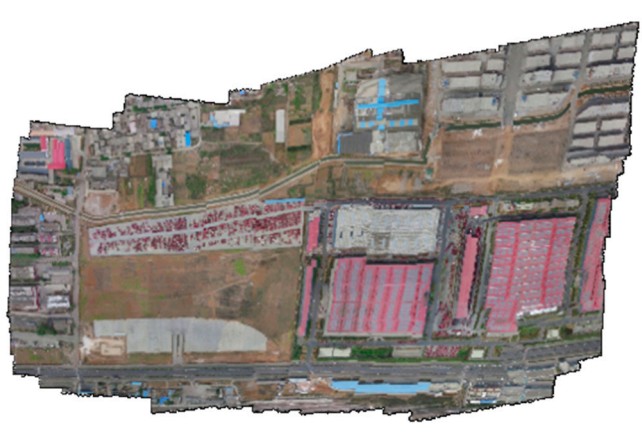

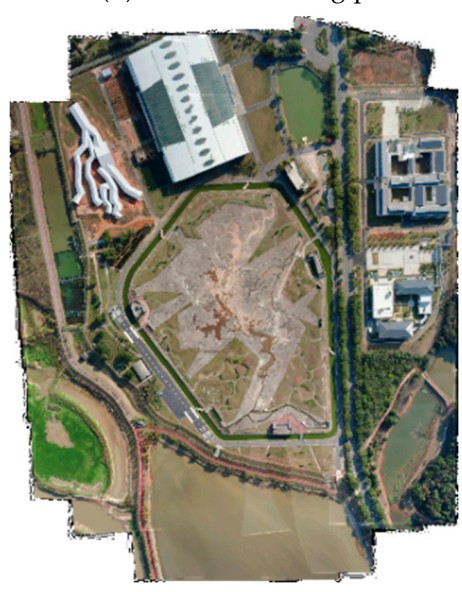

(**c**) Phantom3-factory

(**d**) CW5-boyang

**Figure 5.** UAV image mosaicking result of the proposed system in real scenes. The sequences Phantom3-village, Phantom3-huangqi, and Phantom3-factory are from the NPU drone map dataset and CW5-boyang is collected by us.

Sequence Phantom3-village was taken by a phantom3 UAV in the area of Hengdong, Hunan, and the mosaic results are shown in the first row and first column of Figure 5. The area mainly covers a plain of scattered villages with several running paths. The paths are well mosaicked and the houses are shown in their original shapes. The upper part of the area is adjacent to a piece of water. Its feature points are unstable and difficult to extract, so the position of the calculated image is affected and some gaps on the waterfront are not fully aligned.

Phantom3-huangqi was taken by a phantom3 UAV in the area of Hengdong, Hunan, and the mosaic results can be seen on the right side of the first row of Figure 5. This is mainly a plain area traversed by a highway, interspersed with waters. It covers an area of 1.31 square kilometers, but most of the parts can be regarded as planes, so the

mosaic results of the roads and waters are pretty good. However, the roads at the edge are misaligned. This is because our algorithm eliminates the cumulative error by searching the neighborhood of the current frame and then optimizing it locally, and the road at the edge lacks surrounding supports. However, multiple images in the same strip also inhibit the expansion of this misalignment.

Phantom3-factory was taken by a phantom3 UAV in the area of Luoyang, Henan Province, and the mosaic results are shown on the left side of the second line. This set of images includes some open spaces such as factory buildings and parking lots. There are some misalignments at the edges of some houses because of the shape of the houses and the relief of the terrain. However, thanks to our optimization algorithm, these misalignment errors are not accumulated in the mosaic process. They are just scattered in the image and do not lead to greater misalignment. Moreover, because of our fusion algorithm, strong pixel segmentation does not appear in the images, and in most affected areas, pixel transitions are natural and reasonable.

The last set of the data sequence is obtained by CW-15 UAV in a test site. In this sequence, our UAV kept flying at a low altitude of about 150 m. The area contains an open space surrounded by houses of different sizes, and waters at the edge of the area. Although there are many elements in the image, our algorithm still restores this area well, and the details are restored better because of the low flight altitude. The edge of the white house in the image produces some small pixel misalignment, which is caused by the lack of relevant frames for position settlement when the UAV reaches the edge of the strip. However, the misalignment is corrected by the algorithm in the image mosaic and does not cause greater misalignments.

The experiment shows excellent performance and robustness in the sequences of different ground objects taken by different UAVs at different flight altitudes. The keyframe extraction strategy and local optimization strategy of our algorithm greatly reduces the common error accumulation in multi-strip mosaics, and our fusion algorithm also makes some small mosaic misalignments more natural and improves the overall mosaic performance.

### 3.3. Robustness Experiment

In this section, we focus on the robustness of our algorithm in different scenarios. The robustness experiments consist of two parts: robustness experiments to changes in lighting and noise, and robustness experiments to moving targets. The former tests our algorithm's performance under different weather conditions and images obtained from different sensors through a series of artificial changes of brightness and noise, while the latter examines whether our algorithm is robust to interference from moving targets.

### 3.3.1. Robustness Experiment to Changes in Lighting and Noise

Due to different sensors and weather changes, it is difficult to ensure consistency in image quality, though it is obtained after surveying and mapping the same area. The image quality is usually affected by random noise and brightness changes. However, an excellent algorithm needs to be able to run stably in a variety of situations.

We selected Phantom3-village to process the image sequence. Each image in the sequence is completely random as to whether noise is added, the ratio of noise added, whether brightness changes are made, the region of brightness changes, and the intensity. We use the newly generated data as the input of the algorithm, and compare the speed and quality with the unprocessed mosaic results. The experimental results are shown in Figure 6.

In the mosaic result, many irregular bright blocks and dark blocks are distributed in the image. These blocks affect our feature performance at different scales. However, SIFT extraction algorithms are not sensitive to brightness and noise, and our local optimization algorithm can eliminate those misalignments. As a result, there are no serious misalignments, and the accuracy is not greatly affected compared with the original mosaic results.

However, in some areas, such as in the yellow frame, a few pixel misalignments are shown in the road. The overall mosaic result performs well.

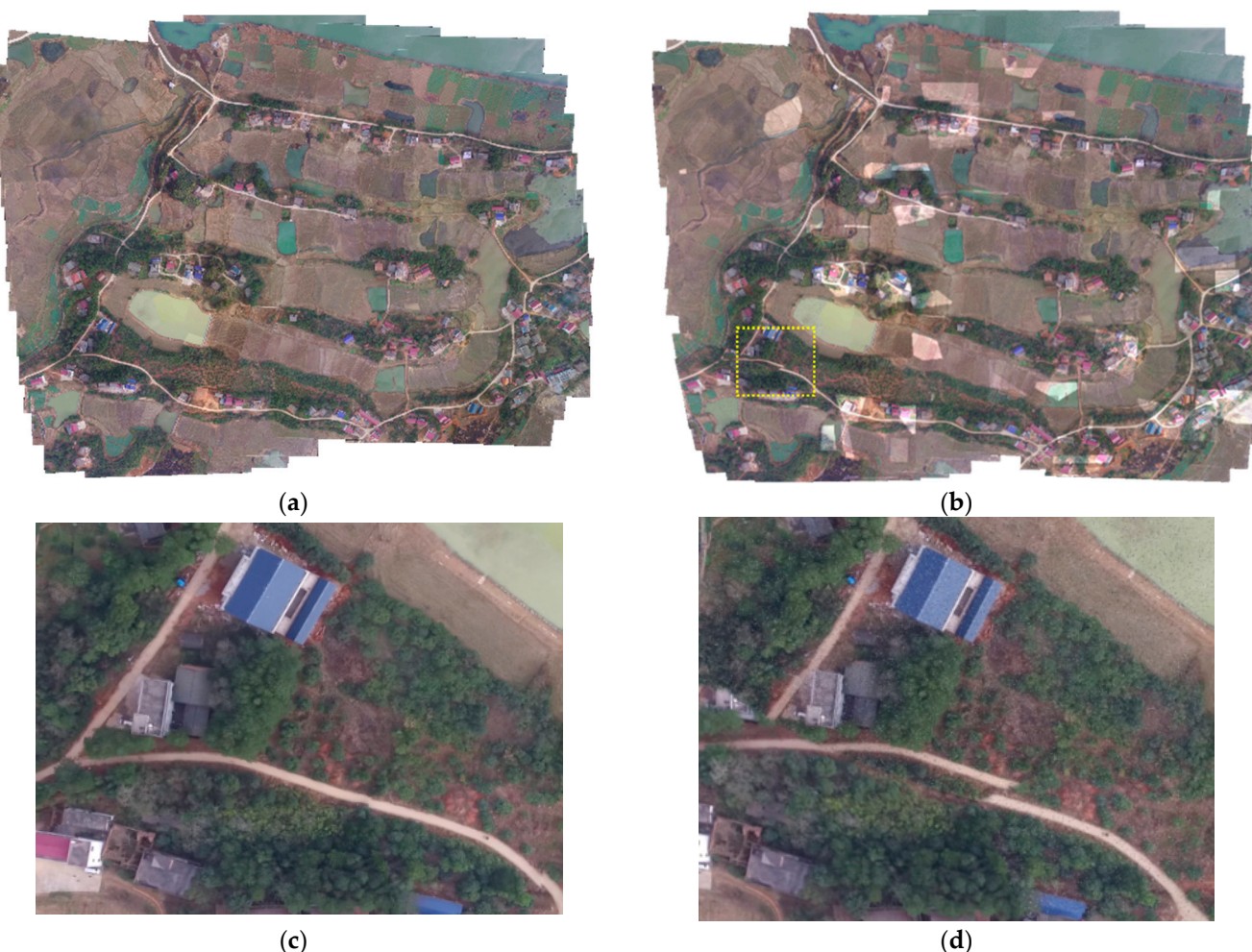

**Figure 6.** Comparison of sequences. Original Phantom3-village (**a**) and the randomly processed one (**b**). Images (**c**,**d**) are the detail of the yellow dash box in panoramas.

To sum up, the proposed image mosaic algorithm is robust to brightness changes. It can avoid image brightness adjustment preprocessing, thus reducing the computational burden.

### 3.3.2. Robustness Experiment of Moving Targets

In practical applications, moving targets often exist in image sequences, which may affect the stitching result. In this study, we use real-world data containing motion targets to verify the robustness of our algorithm in this aspect. We captured aerial imagery of a road using the CW-15 with three flight strips to demonstrate the robustness of our algorithm in the presence of moving targets. The road had many moving vehicles, and the vehicles' positions were not fixed in each frame, making it more challenging than stitching frames without moving targets. We processed the collected data with our algorithm and obtained the following results:

Figure 7a shows the trajectory of the collected data, indicating that we flew three flight lines above the road to ensure multiple captures. Figure 7b is the result of our algorithm, which shows that the panorama is not misaligned due to moving targets such as vehicles. We calculate the position of each frame by computing the transformation of a large number of matching point pairs. In this process, moving vehicles are treated as noise. Unlike the least squares algorithm, which is sensitive to noise, the RANSAC algorithm we used separates all data into inliers and outliers, thereby accurately computing the transformation

while removing noise. Moreover, Figure 7c shows a close-up of the vehicles on the road. Our fusion algorithm ensures that the images of the vehicles on the road are clear without blurring or ghosting, contrary to expectations. More comparative experiments and analyses of fusion algorithms can be found in Section 3.4.6.

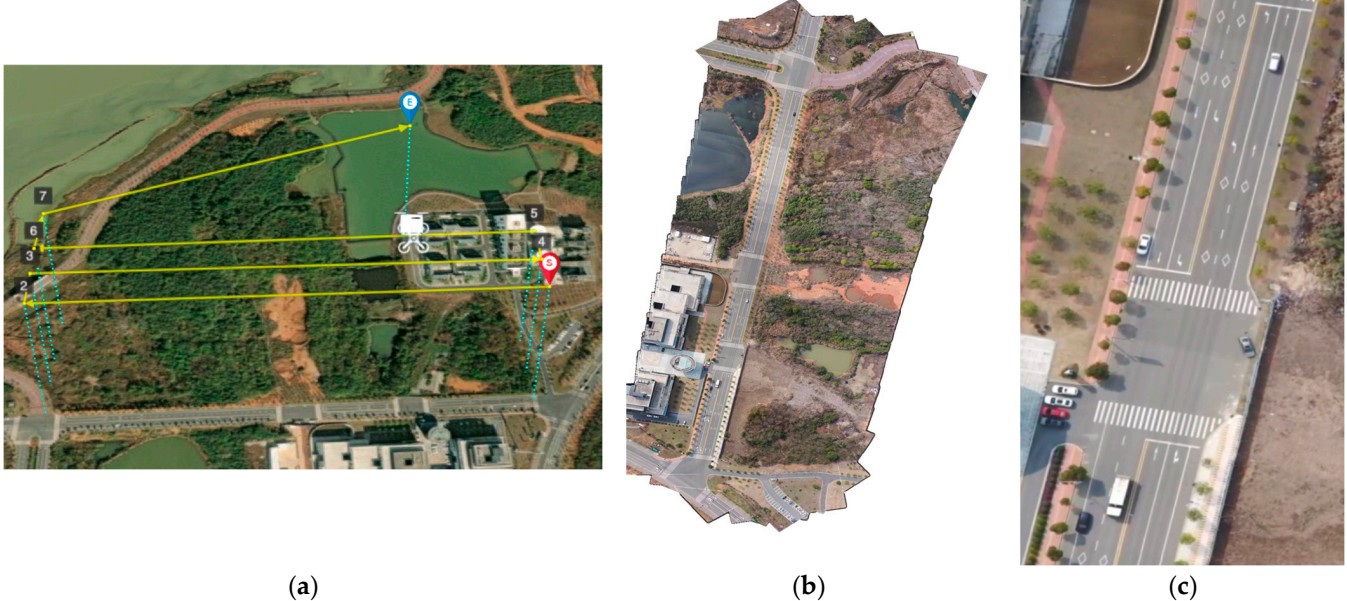

<div align="center">(<b>a</b>)      (<b>b</b>)      (<b>c</b>)</div>

**Figure 7.** Result of robustness experiment of moving targets. Including (**a**) flight trajectory, (**b**) panorama from our framework and (**c**) detailed close-up of the vehicle.

The above experiments and results demonstrate that our algorithm exhibits strong robustness when processing images containing moving targets. It effectively avoids image stitching errors caused by moving targets, and with the help of our fusion algorithm, moving targets in the image can also be correctly displayed in the panoramic image. Therefore, our algorithm has great practical value for processing images in real scenes.

### 3.4. Comparative Experiment

In this section, we designed a series of comparative experiments. The comparison experiment included two parts: comparison of strategies and comparisons with other UAV mosaicking algorithms. The comparison of strategies was conducted to demonstrate the effectiveness of the strategies selected in our framework. We designed four sets of comparative experiments to demonstrate the superiority of the SIFT extraction algorithms we selected over other feature extraction algorithms, as detailed in Sections 3.4.1–3.4.4. In this section, we simulate image transformations in different environments and measure the performance of feature extraction algorithms by matching them using different feature extraction techniques. We visualize these matches, although due to the large number of connecting lines between matched points, it may be difficult to distinguish each line with the naked eye. However, we can judge the overall trend. Generally, better feature extraction algorithms can obtain matching lines with consistent directions. To further illustrate this trend, we have calculated the correct matching rate and processing time of these feature extraction algorithms to quantitatively measure their performance.

We also validated the effectiveness of our keyframe selection strategy through a set of ablation experiments detailed in Section 3.4.5. Finally, we compared our fusion algorithm with other fusion methods to demonstrate its efficacy in Section 3.4.6. In the comparison experiment with other UAV mosaicking algorithms, we validated the performance of our framework in terms of effect and speed by comparing it with mature commercial software

such as QuickBird [35] and AutoPano [36], as well as advanced stitching algorithms such as Open-Stitcher [37].

3.4.1. Comparative Experiment of Feature Extraction Algorithms under Random Noise

Salt and pepper noise is common in UAV images, which is usually caused by pixel failure due to sensor interference or transmission error. We add random salt and pepper noise to images to test the performance of various features. We evaluate our algorithm by matching images with and without special processing using different features, using processing time as an efficiency indicator and match rate as a quality indicator. The experimental results are shown in Figure 8 and Table 3.

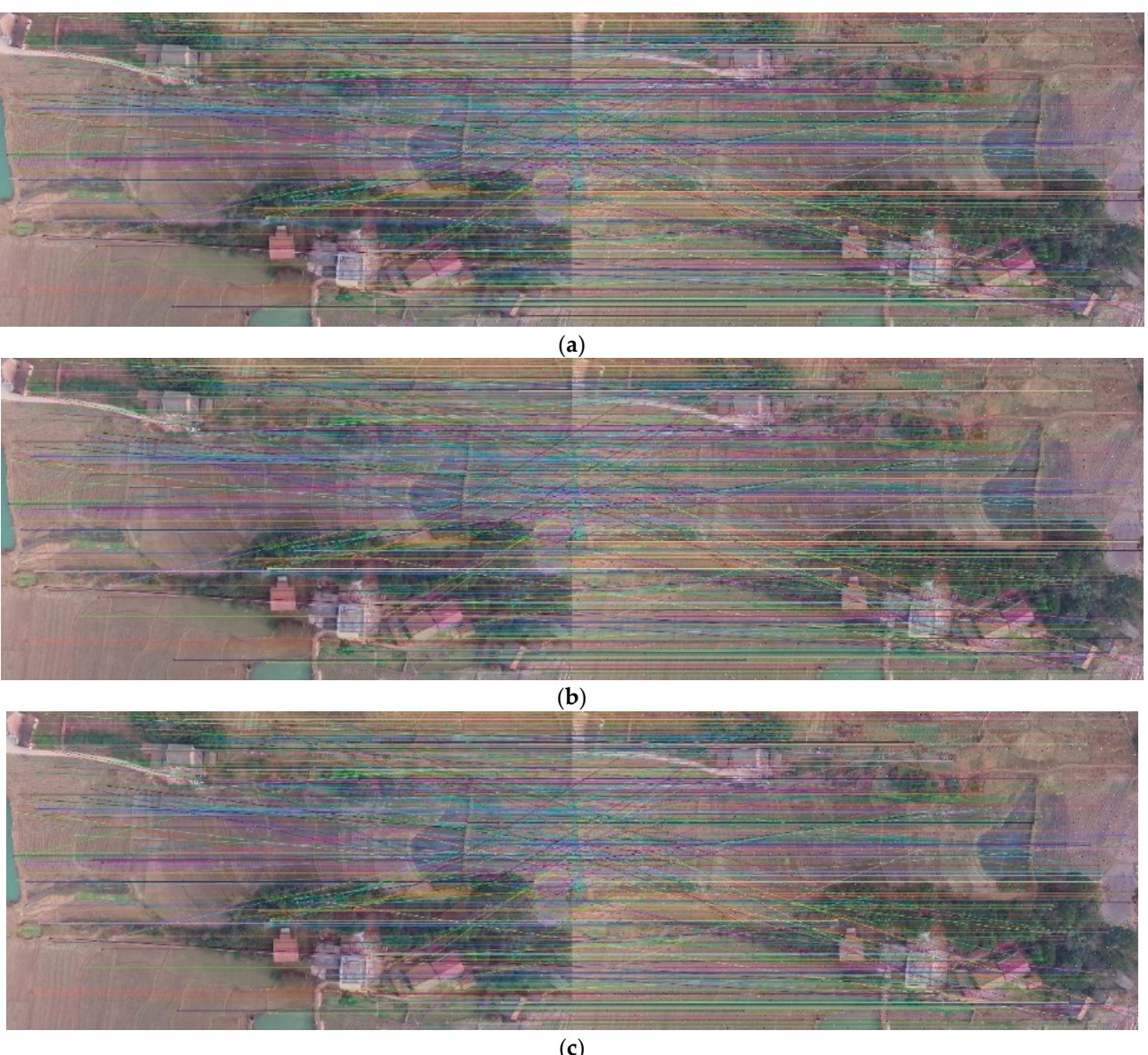

**Figure 8.** The matching of the original image with the image added with a salt and pepper noise using (**a**) SIFT feature algorithm (**b**) SURF feature algorithm (**c**) ORB feature algorithm.

As the results show, among the three features, the SIFT feature extraction algorithm has the highest matching rate, reaching 63%, and is 10% higher than the other two feature extraction algorithms, which are only half of the correct rate. It shows that the SIFT has the strongest resistance to salt and pepper noise and is more robust to different sensors and environments.

**Table 3.** Results of the image matching by adding salt and pepper noise randomly. the image changed brightness. The best result is marked in bold.

| Feature Extraction Algorithm | Time (s) | Matches | Correct Matches | Match Rate |
|---|---|---|---|---|
| SIFT | 0.387 | 7970 | 5036 | **0.632** |
| SURF | 0.502 | 10987 | 5865 | 0.534 |
| ORB | 0.050 | 500 | 251 | 0.502 |

3.4.2. Comparative Experiment of Feature Extraction Algorithms under Random Brightness Change

When the UAV takes aerial photos, due to time and position differences, the image will show different shadows and the local brightness of the image changes. We randomly pick some areas on the image to change the brightness and test the performance of features. The results are shown in Figure 9 and Table 4:

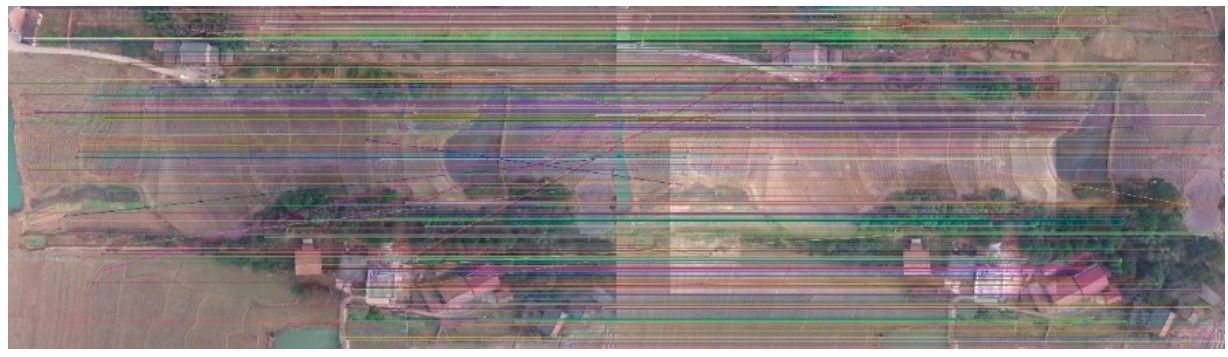

(**a**)

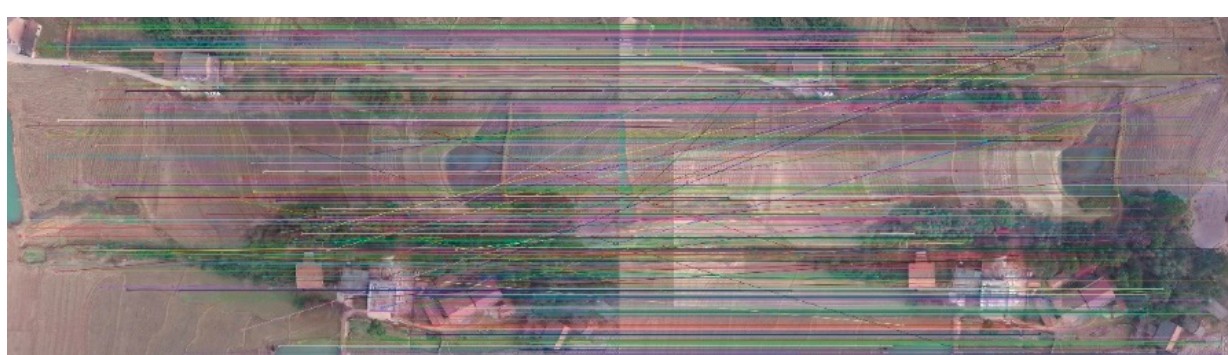

(**b**)

(**c**)

**Figure 9.** The matching of the original image with the image changed brightness using (**a**) SIFT feature extraction algorithm (**b**) SURF feature extraction algorithm (**c**) ORB feature extraction algorithm.

**Table 4.** Results of comparing the image with the image changed in brightness. The best result is marked in bold.

| Feature Extraction Algorithm | Time (s) | Matches | Correct Matches | Match Rate |
|---|---|---|---|---|
| SIFT | 0.368 | 7970 | 7857 | **0.986** |
| SURF | 0.524 | 10987 | 10678 | 0.972 |
| ORB | 0.042 | 500 | 381 | 0.762 |

From the result, the SIFT feature extraction algorithm has the highest matching rate of 98%, while the ORB has the lowest. Thus, the SIFT feature extraction algorithm has better robustness under aerial survey tasks with more brightness changes.

### 3.4.3. Comparative Experiment of Feature Extraction Algorithm under Random Rotation

The angle change of the image is very common in UAV mapping. This geometric transformation leads to the absolute orientation change of the image. We tested the rotation of 45 degrees, and the results are shown in Figure 10 and Table 5.

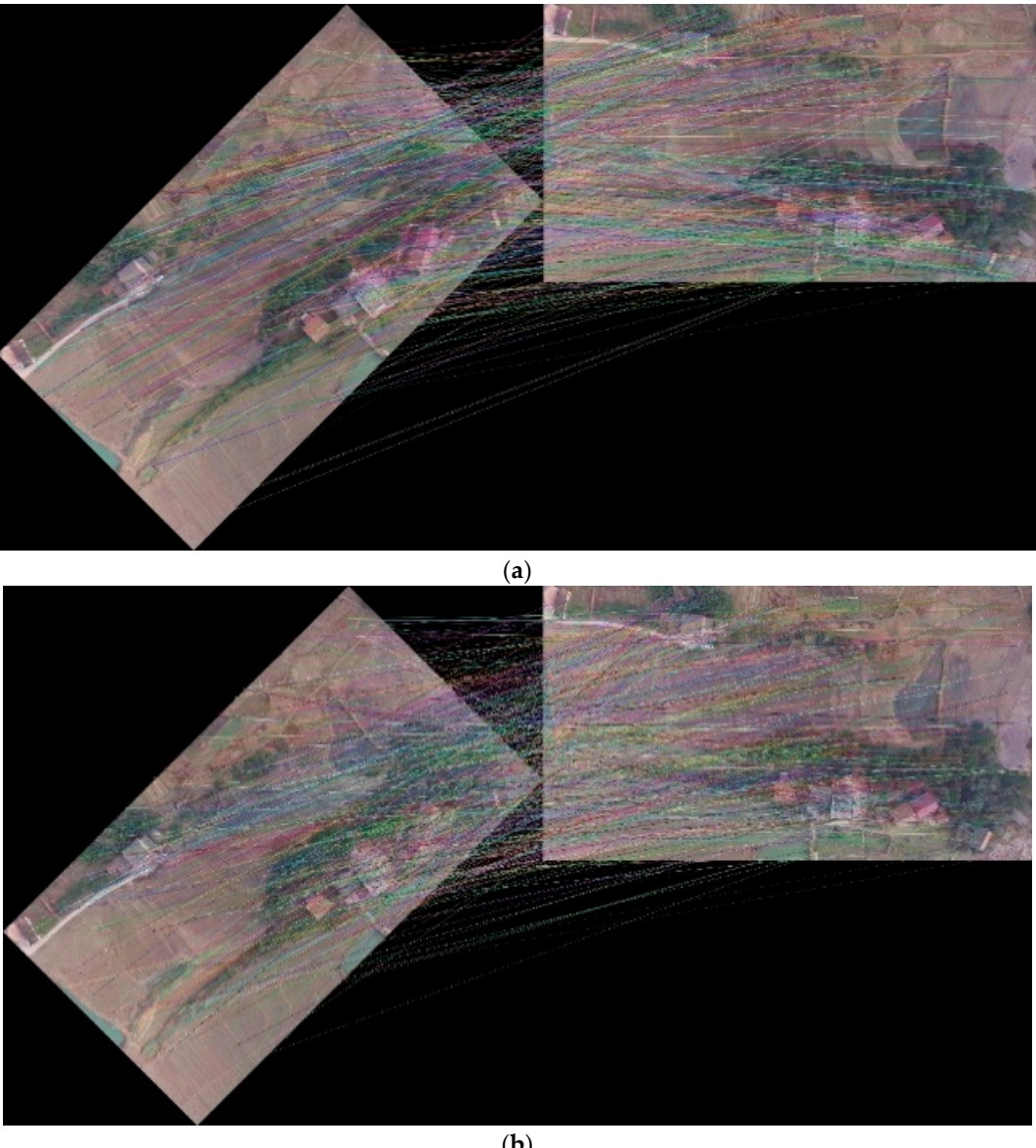

(**a**)

(**b**)

**Figure 10.** *Cont*.

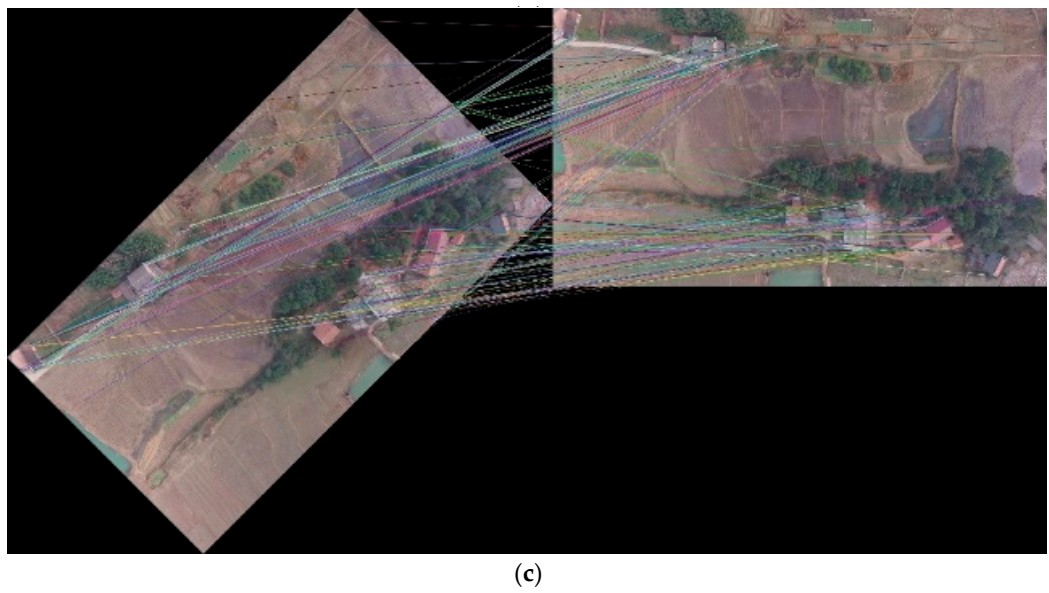

(**c**)

**Figure 10.** The matching of the original image with its rotated image using: (**a**) SIFT feature extraction algorithm (**b**) SURF feature extraction algorithm (**c**) ORB feature extraction algorithm.

**Table 5.** Results of comparing the image with its rotated image. The best result is marked in bold.

| Feature Extraction Algorithm | Time (s) | Matches | Correct Matches | Match Rate |
|---|---|---|---|---|
| SIFT | 0.731 | 7544 | 4846 | 0.642 |
| SURF | 0.636 | 10954 | 4870 | 0.445 |
| ORB | 0.084 | 500 | 325 | **0.650** |

The ORB feature extraction algorithm has a correct matching rate of 65%, the SIFT feature extraction algorithm is almost the same as the ORB feature extraction algorithm, and the SURF feature extraction algorithm is less resistant to rotation. As shown in Table 6, we compare the performance of each feature extraction algorithms at 45 degree intervals. The result shows that the SIFT feature extraction algorithm has the best matching rate when the rotation angle is a multiple of 90, and the ORB feature extraction algorithm has the best matching rate in other cases, but the difference between the SIFT feature extraction algorithm and ORB feature extraction algorithm matching rate is small in these cases, while SURF has the lowest matching rate at all angles.

**Table 6.** Matching rate versus the rotation angle. Best results are marked in bold.

| Feature Extraction Algorithm | 45° | 90° | 135° | 180° | 225° | 270° |
|---|---|---|---|---|---|---|
| SIFT | 0.642 | **0.966** | 0.638 | **0.955** | 0.639 | **0.970** |
| SURF | 0.445 | 0.962 | 0.444 | 0.951 | 0.445 | 0.963 |
| ORB | **0.650** | 0.916 | **0.666** | 0.880 | **0.674** | 0.918 |

### 3.4.4. Comparative Experiment of Feature Extraction Algorithms under Scale Change

During the flight of the UAV, altitude changes leads to a scale change of the same ground area in the image. We randomly changed the scale and tested the performance of features. The results are shown in Figure 11 and Table 7.

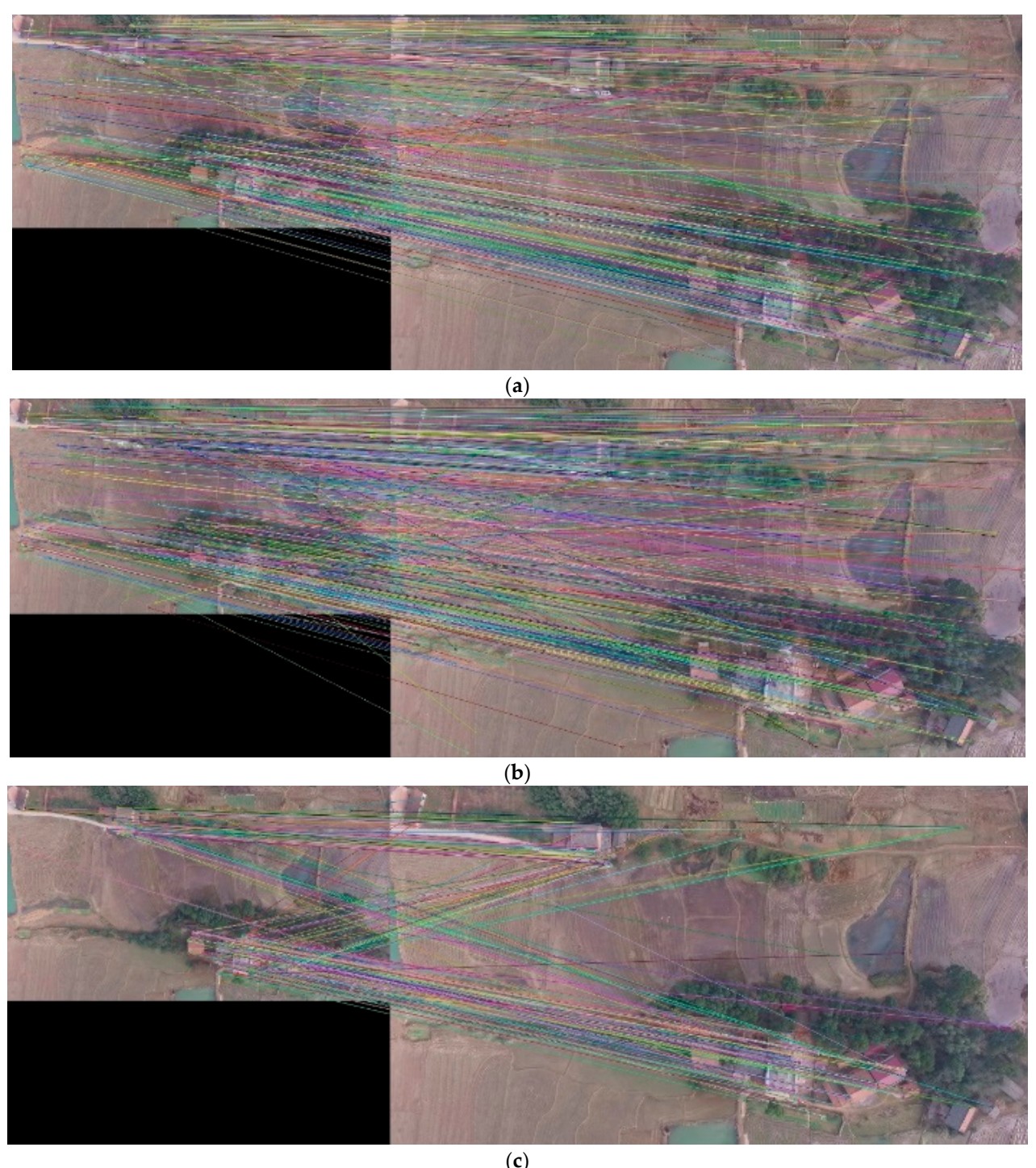

**Figure 11.** The matching of the original image with its scaled image using: (**a**) SIFT feature extraction algorithm (**b**) SURF feature extraction algorithm (**c**) ORB feature extraction algorithm.

**Table 7.** Results of comparing the image with its scaled image. The best result is marked in bold.

| Feature Extraction Algorithm | Time (s) | Matches | Correct Matches | Accuracy |
|:---:|:---:|:---:|:---:|:---:|
| SIFT | 0.249 | 1765 | 3530 | **0.751** |
| SURF | 0.393 | 4254 | 8508 | 0.721 |
| ORB | 0.032 | 500 | 273 | 0.546 |

The SIFT feature extraction algorithm has the highest matching rate in scale change and performs much better than the ORB feature extraction algorithm. The SIFT feature extraction algorithm has high robustness when UAV flight altitude changes, so it can still get excellent matches when the scale changes.

### 3.4.5. Comparison of Keyframe Selection Strategies

The aerial sequence acquired by the UAV has a lot of images. In the mosaic process, we do not regard every image acquired as the keyframe needed for final mapping. Sparse selection is easy to fail, and full frame mosaicking will make a lot of redundant calculations. Such calculations do not improve the mapping results much, but the computational resources consumed are huge. We designed an experiment to prove the effectiveness of our keyframe selection algorithm and we adopted the phantom3-village sequence. First, we do not use any selection strategies, and all images participate in settlement for mosaicking, timing, and storing results. Then, we use our keyframe selection strategy to compare their performance and efficiency. The results are as Table 8 and Figure 12.

**Table 8.** Comparison of full frame mosaicking and keyframe selection mosaicking.

|  | Full Frame Mosaicking | Keyframe Selection Mosaicking |
|---|---|---|
| Adopt frames/all frames | 406/406 | 118/406 |
| Time consumed (s) | 384 | 77 |

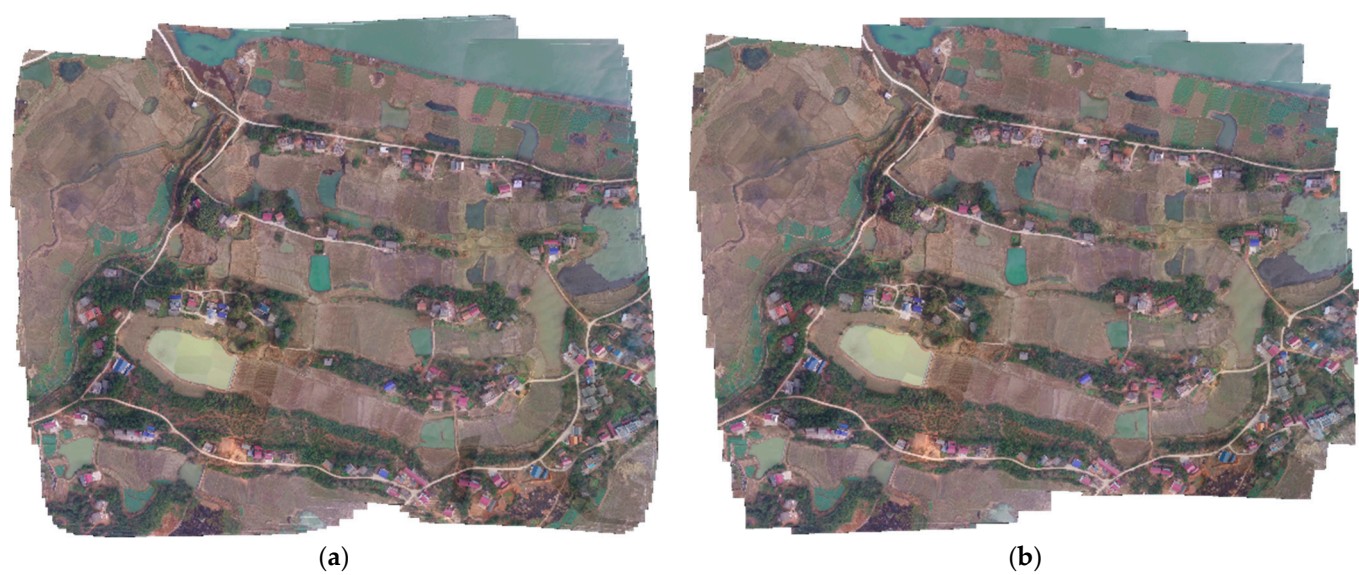

(**a**)       (**b**)

**Figure 12.** Comparison of full frame mosaicking and keyframe selection mosaicking. (**a**) Full frame mosaicking. (**b**) Keyframe selection mosaicking.

As shown in Figure 12, both strategies have successfully obtained panoramas in multi-strip aerial survey missions. A total of 118 frames from this set of data are keyframes. Our algorithm selects 1/4 frames for subsequent solution and mosaicking. The mosaic results of our algorithm do not show obvious pixel misalignment and mismatching. Thanks to the fact that we use fewer frames to solve the problem, our algorithm only takes 77 s, which is equivalent to 1/5 of the algorithm without keyframe selection, which enables our algorithm to mosaic images in real-time during the data transmission from the UAV. However, we discarded a part of the image at the edge, so the edges of the images are coarser than the algorithm without selecting keyframes. However, in the aerial survey mission, the number and length of strips set by the UAV include redundant observations at the edges of the survey area. Under this premise, our algorithm will not lose the information on the survey area.

### 3.4.6. Comparison of Fusion Methods

Fusion is an important step in panoramic image generation. However, the difference in camera exposure, incomplete alignment geometric transformation, and the simple superposition of two images will lead to obvious visual disharmony. That is why an excellent fusion algorithm is needed. For two images, we tend to focus on their fusion effect at the seam and the inconsistent pixels. We compare four fusion methods:

1. Simple coverage, that is, after aligning two images, pixel coverage is performed.
2. Weighted fusion. For the overlapping parts of two images, we weigh the pixel values of both to calculate the new pixel values.
3. Weight substitution. For each image, we construct a mask to express the priority value of the pixel, which represents the position of the pixel near the center of the image. Then, we compare the mask of the two images to decide which pixel to choose.
4. Our method.

The stitching results are shown in Figure 13, and the result of seam and house in the panorama are shown in Figures 14 and 15. After simple coverage, the seam is serious with no algorithm processing. The ground and the house do not show their complete shapes. With the weighted fusion algorithm, the seam is not that serious, but houses are ghosting. The effect of the weight substitution algorithm is similar to that of direct substitution, but in multi-strip sequence images, the weight replacement algorithm can make the obtained panorama more similar to the result obtained after shooting vertically down. Finally, our method, with the help of the Gaussian pyramid, fuses in multiple scales to eliminate stiff seams and preserves the details of the house well without ghosting.

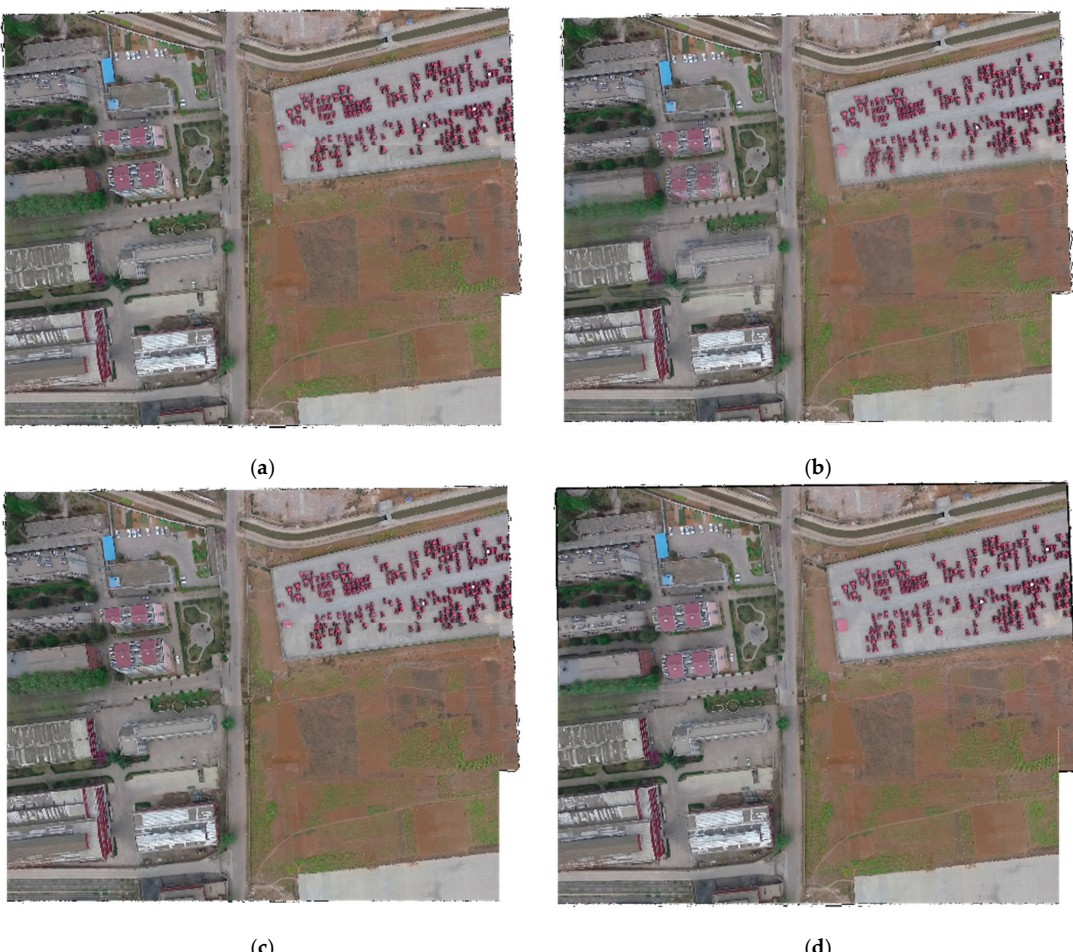

(a)  (b)

(c)  (d)

**Figure 13.** Compositing comparison using different fusion methods. (**a**) Simple coverage. (**b**) Weighted fusion. (**c**) Weight substitution. (**d**) Our method.

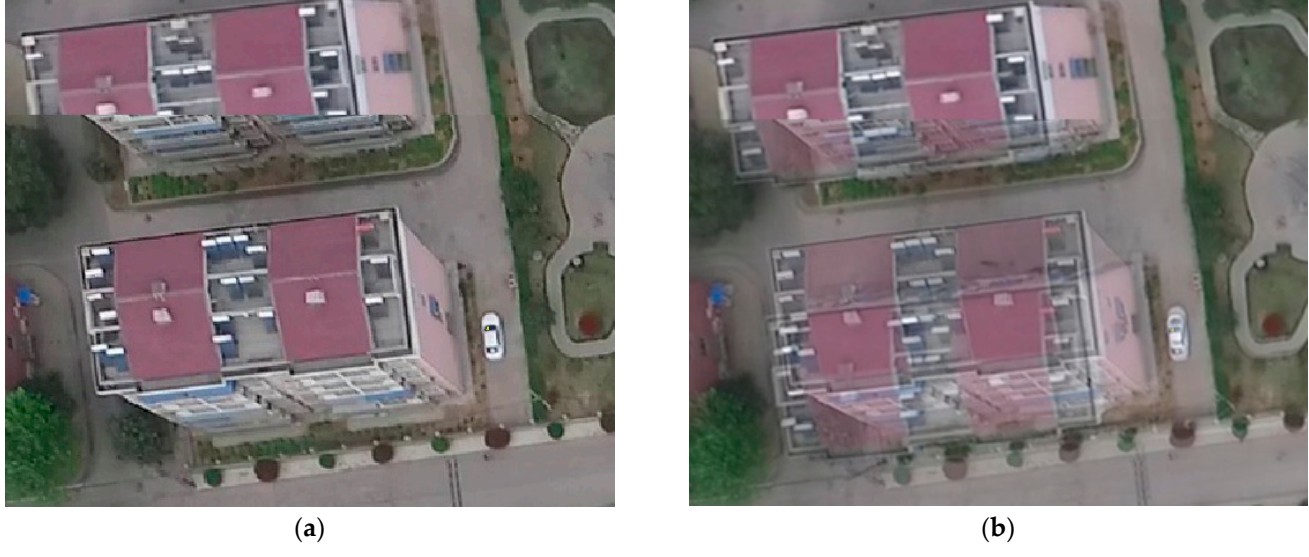

**Figure 14.** Compositing comparison of seam in panorama using different fusion methods. (**a**) Simple coverage. (**b**) Weighted fusion. (**c**) Weight substitution. (**d**) Our method.

**Figure 15.** *Cont.*

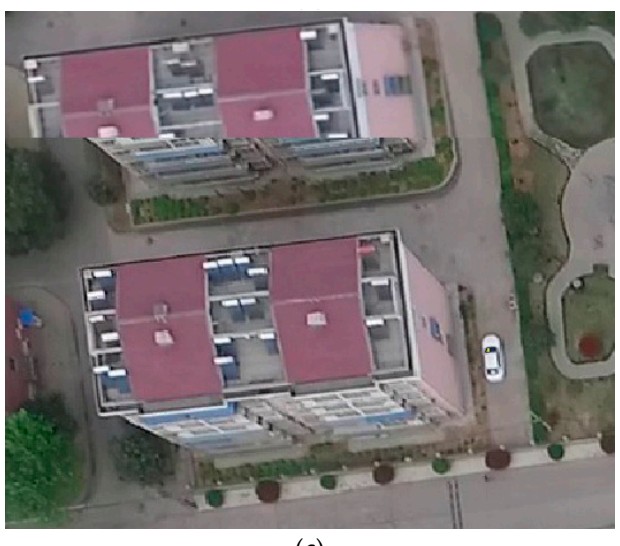
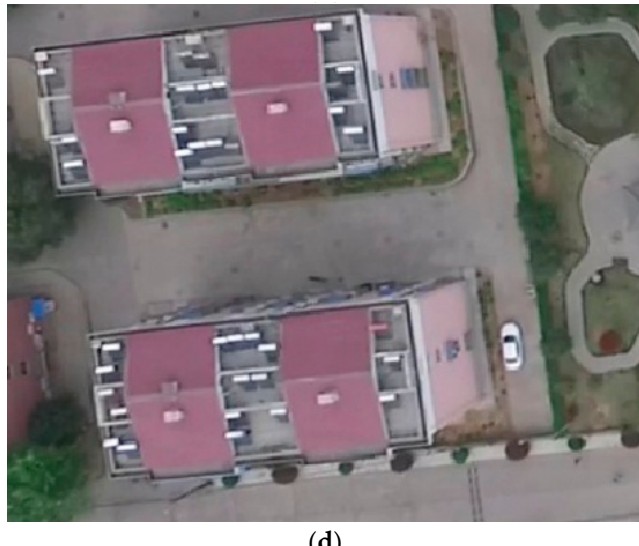

(**c**)                                                                                                                    (**d**)

**Figure 15.** Compositing comparison of house in panorama using different fusion methods. (**a**) Simple coverage. (**b**) Weighted fusion. (**c**) Weight substitution. (**d**) Our method.

### 3.4.7. Comparison with Other UAV Mosaicking Algorithms

In our experiments, we compared QuickBird [35] and AutoPano [36], two pieces of well-established commercial stitching software, with Opencv-Stitcher, an advanced stitching algorithm. Our experiments were conducted on the Phantom3-village, and the results of each algorithm are presented in Figure 16. Additionally, Figure 17 displays the details of the roads, while Figure 18 highlights the details of the houses. In order to compare the efficiency of the algorithm, we recorded the algorithm time consumed, which can be seen in Table 9.

**Table 9.** Comparison of time consuming.

|  | QuickBird | AutoPano | Opencv-Stitcher | Our Method |
|---|---|---|---|---|
| **Time consumed** | 1 min, 11 s | 48 min, 04 s | Stitch-failure | 1 min, 17 s |

Quick-Bird is a real-time processing system for dynamic drone videos. It is a real-time map stitching software developed independently by company ZTmapper [35], which solves another major problem in photogrammetry and fills a long-standing gap in this field both domestically and abroad. Quick-Bird is an unmanned aerial video processing system that integrates advantages such as a fully automatic processing process, real-time keyframe extraction and stitching, support for target detection, and massive data processing capabilities. Autopano is a product of the virtual reality software company Kolor [36]. It is a powerful, easy-to-use, practical, and professional image stitching software that supports more than 400 input file formats. Opencv-stitcher is a new module added in OpenCV 2.4.0 [37]. Its function is to achieve image stitching. The algorithm has many parameters, including feature point categories, coordinate transformation models, and so on. In the experiment, the parameters were set as default.

We observed that QuickBird, AutoPano and our method successfully stitched together the panorama, while Opencv-Stitcher failed to do so. Particularly, in Figure 17, we can see a road that spans two flight strips, which QuickBird and AutoPano were unable to stitch successfully, but our method was able to. This is because our method greatly suppresses the cumulative error caused by dozens of images between two flight strips. Additionally, we observed that when dealing with the breakage of roads, QuickBird employed a rough fusion method, which made the breakage very abrupt, while AutoPano used a weighted fusion method to handle visual errors, but this method made the broken road very blurry.

In contrast, our method can better preserve the structure of the road and does not produce obvious breakage or blurring.

(**a**)

(**b**)

Stitching failure

(**c**)

(**d**)

**Figure 16.** Comparison panorama using different methods. (**a**) QuickBird. (**b**) AutoPano. (**c**) Opencv-Stitcher. (**d**) Our method.

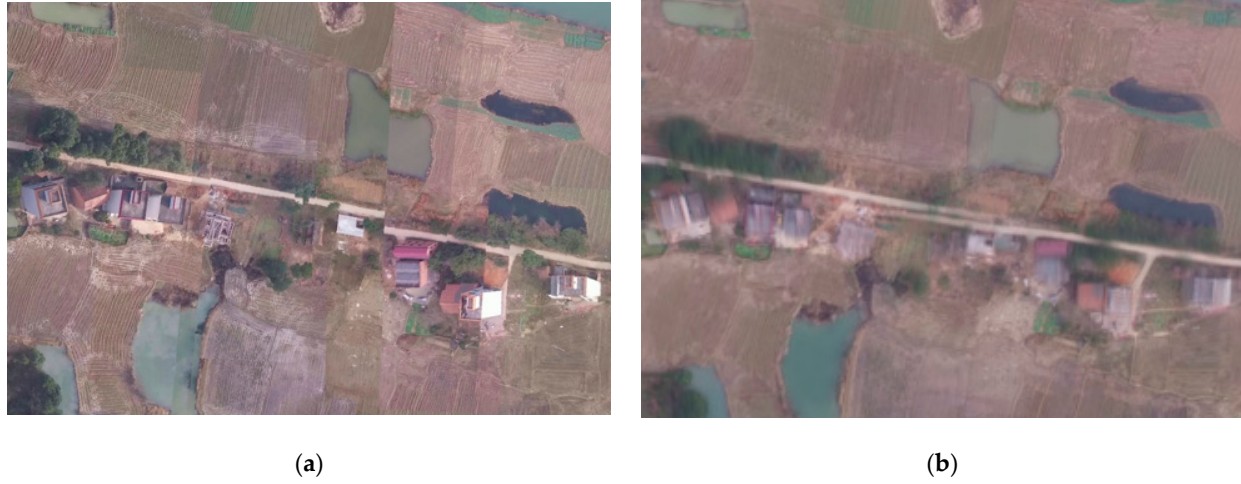

(**a**)

(**b**)

**Figure 17.** *Cont.*

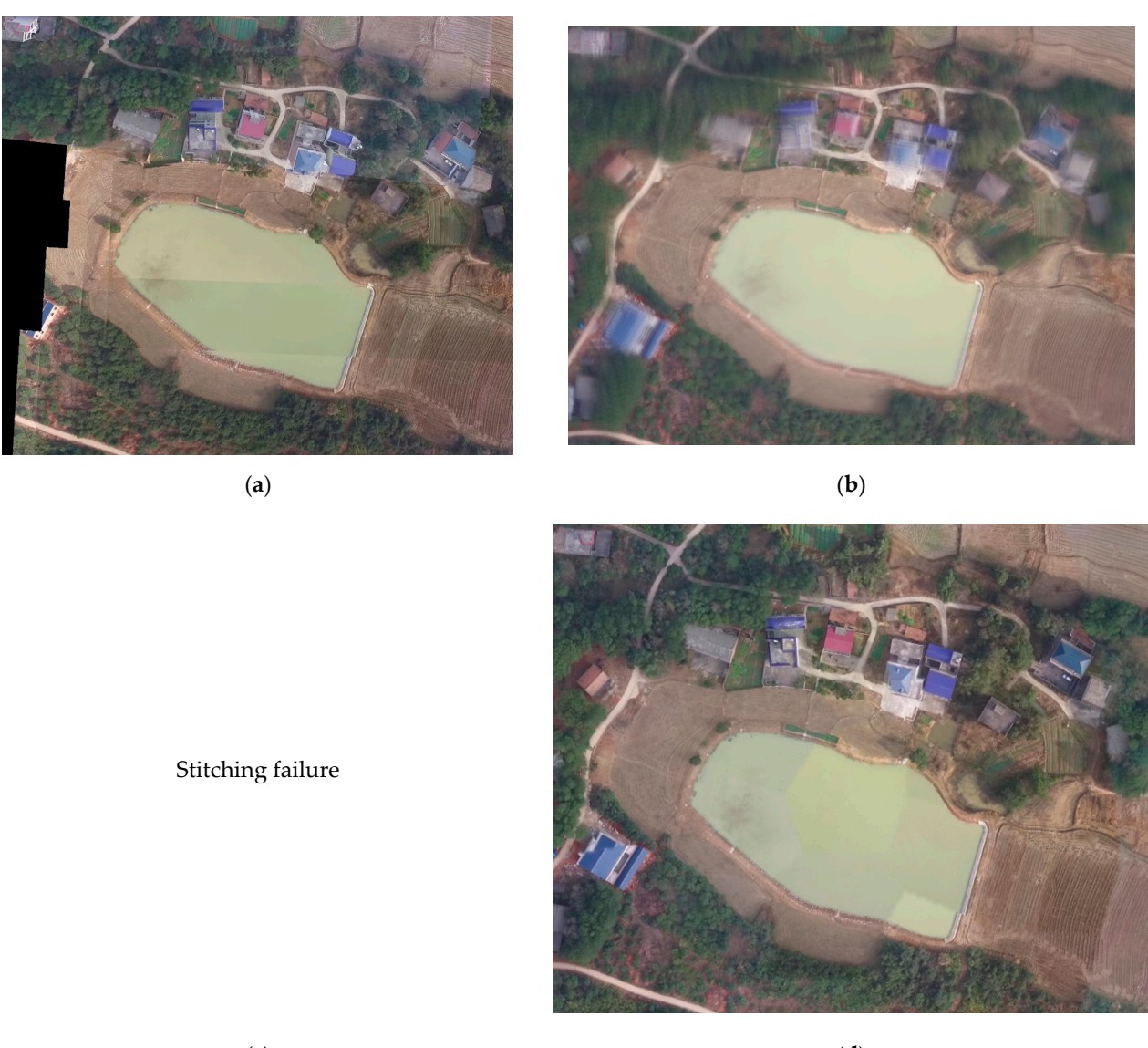

Stitching failure

(**c**)

(**d**)

**Figure 17.** Comparison of road in panorama using different methods. (**a**) QuickBird. (**b**) AutoPano. (**c**) Opencv-Stitcher. (**d**) Our method.

(**a**)

(**b**)

Stitching failure

(**c**)

(**d**)

**Figure 18.** Comparison of house in panorama using different methods. (**a**) QuickBird. (**b**) AutoPano. (**c**) Opencv-Stitcher. (**d**) Our method.

In Figure 18, we show the details of houses. Due to the height differences between buildings and the ground, even with perfect registration, pixel displacement may occur. Therefore, an efficient fusion algorithm is needed to eliminate this phenomenon. We found that QuickBird still used a rough fusion method here, resulting in some buildings being directly truncated, while AutoPano's result manifested as many, very blurry buildings. Compared to this, our method can display the structure of the buildings clearly and does not produce truncation or blurring phenomena.

After comparing the processing times of different algorithms, our algorithm was able to complete the stitching of 406 images in the Phantom3-village dataset with a resolution of 1920*1080 within 1 min and 17 s. This processing time is similar to that of the QuickBird algorithm, while the AutoPano algorithm took over 48 min to generate a panorama of this dataset.

The Phantom3-village dataset covers an area of 0.9 square kilometers, and it typically takes several minutes for a drone to capture the necessary images. Our algorithm is capable of real-time parallel stitching during the process of image transmission from the drone.

Therefore, by comparing our method with QuickBird and AutoPano, we found that our method performs better in stitching panoramic images and can better handle visual errors and fusion problems. This proves the effectiveness and feasibility of our method in UAV image mosaickin.

## 4. Discussion

Through experiments on public datasets and our collected data, we verify the effectiveness of our algorithm. Our algorithm can output high-quality results in various ground scenes, and remains robust under the influence of varying lighting conditions and sensor noise. Additionally, we tested the keyframe selection strategy and the weighted partition fusion method based on the Laplacian pyramid. Experiments show that our keyframe selection strategy can greatly accelerate the stitching speed without affecting the result quality so that our algorithm can synchronize when the UAV executes the task. Compared to other methods, our fusion method not only preserves the complete form of ground objects such as tall buildings, but also avoids obvious ghosting and stiff seams, so as to output better panorama.

## 5. Conclusions

In this paper, we propose a real-time incremental UAV image mosaicking framework, which only uses the UAV image sequence. The results suggest that our proposed real-time incremental UAV image mosaicking framework shows promising performance compared to other existing methods. Our keyframe selection strategy can greatly accelerate the speed of mosaicking. Moreover, we introduce frame fuzzy positioning and new local optimization strategy, which minimizes the parallax and cumulative error of large sequence images. The fusion algorithm we adopt also makes the algorithm result better.

While our algorithm has promising applications in UAV mapping, emergency management and other fields, it is important to consider its limitations when applying it in real-world scenarios. Our algorithm is aimed at tasks that require fast generation of large panoramic images, such as post-disaster assessments and search and rescue operations. In these tasks, the UAV often needs to fly at higher altitudes to cover larger areas. However, flying at too-low altitudes can increase the impact of dynamic objects and can potentially violate the homography assumption between images. This limitation should be taken into consideration when applying our algorithm in real-world scenarios.

In the future, we will improve the local optimization strategy, further reduce the cumulative error, and find a better algorithm to optimize our algorithm under the condition of GPS. We will also focus on expanding the scope of its applications. Specifically, we aim to apply our algorithm to a wider range of tasks, such as urban planning and infrastructure inspection. To achieve this, we will improve the robustness and scalability of our algorithm, allowing it to handle larger and more complex datasets.

**Author Contributions:** Conceptualization, R.L. and P.G.; methodology, R.L. and X.C. (Xiangyuan Cai); software, R.L. and X.C. (Xiaotong Chen); validation, J.W. and Y.C.; writing—original draft preparation, R.L. and P.G.; writing—review and editing, R.L. and H.Z.; supervision, H.Z.; funding acquisition, H.Z. All authors have read and agreed to the published version of the manuscript.

**Funding:** This research was funded by the National Key Research and Development Program of China, grant number No. 2022YFF0904403 and the National Natural Science Foundation of China, grant number No. 42130104.

**Data Availability Statement:** The data presented in this study are available upon request from the corresponding author.

**Conflicts of Interest:** The authors declare no conflict of interest.

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
