# Peer review of "A Real-Time Incremental Video Mosaic Framework for UAV Remote Sensing"

_remotesensing, doi:10.3390/rs15082127_

Round 1

Reviewer 1 Report (Previous Reviewer 2)

The author has addressed the issues and suggestions related to my last review. I hava no other questions.

Author Response

Your insightful comments have helped us improve our paper significantly. We appreciate your attention to detail and your commitment to helping us make our paper stronger. We have learned a great deal from your feedback and have incorporated your suggestions to enhance the clarity and accuracy of our paper. Thank you once again for your contribution to our work.

Reviewer 2 Report (New Reviewer)

Please see the attached document for more details.

Author Response

Please see the attached document for more details.

This manuscript is a resubmission of an earlier submission. The following is a list of the peer review reports and author responses from that submission.

Round 1

Reviewer 1 Report

Some comments:

1. What the key new points of this manuscript?

2. Is there some experiments carried out with moving targets in the frames?

3. What about the proposed method applied in urban area, mountain area or other areas with complex topography?

4. In the experiments, parameters of UAV and cameras should be described.

5. What is the “accuracy” mean in Table 2?

6. Other quantitative comparison should be considered beside time-consuming.

Reviewer 2 Report

This study proposed a real-time incremental UAV image mosaicking framework, which only used the UAV image sequence. The methods inclueded the keyframes selection, the matching relationship obtaining, the optimization method based on minimizing weighted reprojection errors and the weighted partition fusion method based on Laplacian pyramid. It is interesting and helpful for this topic research and future study.

However, there are some issues or questions could be addressed:

1) In the abstract, the meaning and the important findings of this study should be explained clearly. The meaning of real-time incremental UAV image mosaicking framework proposed by the authos should be explained in detail.

2) In the Section of Introduction, the existing methods such as deep learning-based image mosaicking algorithms were described. But, the problems of previous studies are not described or compared with the proposed method in this study. There is not the explanation of the detailed problems or limitations in these methods. 

3) In this study, the authors should compared the proposed method with the existed popular algorithms, such as ICP, SIFT, or deep learning-based method.

4) In Section 2 of Materials and Methods, the Figure 1 was too simple to explain the proposed framework in detail. The description in Figure 1 was also not  in accordance with the names of different parts.

5) Is the Formula (8) correct given the confusion of the signs? Same as Formula (11).

6) In Section 3.1 of Real-scene experiment, the experiment results are only obtained by the proposed method, which should compared with other existed methods.

7) In Second 3.2 of Robustness experiment, it is difficult to understand the experiment design. The experimental results of Figure 6 are also confused. What is the compared and conditions of this experiment.

8) In Section 3.3, there are different contrast experiment. So, what are the differences between Section 3.2 and Section 3.3? 

9) In Section 3.3, the contrast experiment should placed in Section 4 of Discussion.

10) There are some other issues should be addressed:

a) In Section 3.3, the Table 2 to Table 7 may not be centered. The authors should check carefully;

b) Some figures of results should be represented in detail.

c) Abbreviations that appear for the first time in the content need to be explained clearly., etc. 

Reviewer 3 Report

Comments on Remote Sensing-2220234

The manuscript is unintelligible due to substandard writing and technical faults. The problems of this manuscript are numerous, including, but not limited to, the following cases.

Writing problems:

1.     Inadequate sentences: Try not to use negative wording in a positive mood, see Lines 14-16. It is grammatically fine, but awkward and misleading.

2.     Troublesome abstract and abbreviations: In preparing the abstract, be aware that the abstract is a stand-alone section and the use of abbreviations should be very careful. Refer to any good writers’ handbook to learn how to prepare an acceptable abstract. Also, some acronyms are never explained, it shouldn’t occur in a decent paper.

3.     Lacking required citations: All equations should be precisely cited if those are not the authors’ contributions.

Technical issues:

1.     The authors applied the minimizing weighted reprojection errors to perform precise position calculation, however, my understanding is that the optimization is carried out pixel by pixel. How the authors ensure the smoothness of the surfaces while improving the accuracy. I suggest the authors provide some details of the optimization cost function.

2.     The occlusion problem is crucial in this study, but the authors seem to ignore this issue.

3.     The explanations of most figures and Tables are inadequate. For instance, Figures 7-10 and Tables 4-6 compared the results of three algorithms, SIFT, SURF, and ORB, unfortunately, the authors are unsure what they are. They call them “features” or “feature descriptors”, and even worse, left them “anonymous” (see Tables 4-6). Algorithms are algorithms, not features! In addition, I don’t visually get a significant difference in the results derived from those three algorithms. I suggest the authors provide some hints for readers.